# Sample, Align, Synthesize: Graph-Based Response Synthesis with ConGrS

## Abstract

Language models can be sampled multiple times to access the distribution underlying their responses, but existing methods cannot efficiently synthesize rich epistemic signals across different long-form responses. We introduce Consensus Graphs (ConGrS), a flexible DAG-based data structure that represents shared information, as well as semantic variation in a set of sampled LM responses to the same prompt. We construct ConGrS using a light-weight lexical sequence alignment algorithm from bioinformatics, supplemented by the targeted usage of a secondary LM judge. Further, we design task-dependent decoding methods to synthesize a single, final response from our ConGrS data structure. Our experiments show that synthesizing responses from ConGrS improves factual precision on two biography generation tasks by up to 31% over an average response and reduces reliance on LM judges by more than 80% compared to other methods. We also use ConGrS for three refusal-based tasks requiring abstention on unanswerable queries and find that abstention rate is increased by up to 56%. We apply our approach to the MATH and AIME reasoning tasks and find an improvement over self-verification and majority vote baselines by up to 6 points of accuracy. We show that ConGrS provide a flexible method for capturing variation in LM responses and using the epistemic signals provided by response variation to synthesize more effective responses.

## 1 Introduction

Language models provide interfaces to vast collections of human knowledge. Analogous to information resources such as the internet and the library, LMs are a source of *testimony*: claims distilled from prior sources (Goldman, 2009; Fallis, 2006). Epistemological frameworks suggest that reliable knowledge can emerge from the aggregation of multiple testimonies—agreement between testimonies signals reliability, and disagreement signals potential unreliability, warranting further scrutiny, or complementary perspectives (Fallis, 2006). While a book's credibility can be evaluated through author and publisher identity, citation patterns, and writing style, LM-generated testimony lacks such metadata, posing an epistemic challenge. Unlike static information resources, LMs can produce varied responses per query; we argue that this variation between sampled responses can serve as a valuable epistemic signal, especially when no other external metadata is available. In this paper, we show that LMs can address this epistemic challenge by aggregating over their underlying *distribution* of testimonial claims.

Existing methods to aggregate across LM responses typically take the form of simple majority voting schemes over final short answers (or labels) (Wang et al., 2023b) or selecting a single best response (Bertsch et al., 2023; Chen et al., 2024c). Both types of approaches discard valuable information, and are not generalizable to long-form generation where no one response may contain all the required information. Aggregation methods for long-form generation often incur high costs by relying heavily on LMs to lexically decompose responses into smaller units (Jiang et al., 2024; Wang et al., 2024a;b; Elaraby et al., 2023). Recent progress in reasoning models provides an alternative serial approach, where a single long response (Jaech et al., 2024; Guo et al., 2025) claims to extract more information from the underlying LM, although it requires special training and a very high inference cost, generating many redundant tokens (Chen et al., 2024b). Thus, we ask: How can we *efficiently* synthesize information from multiple LM responses for reliable epistemic signals?

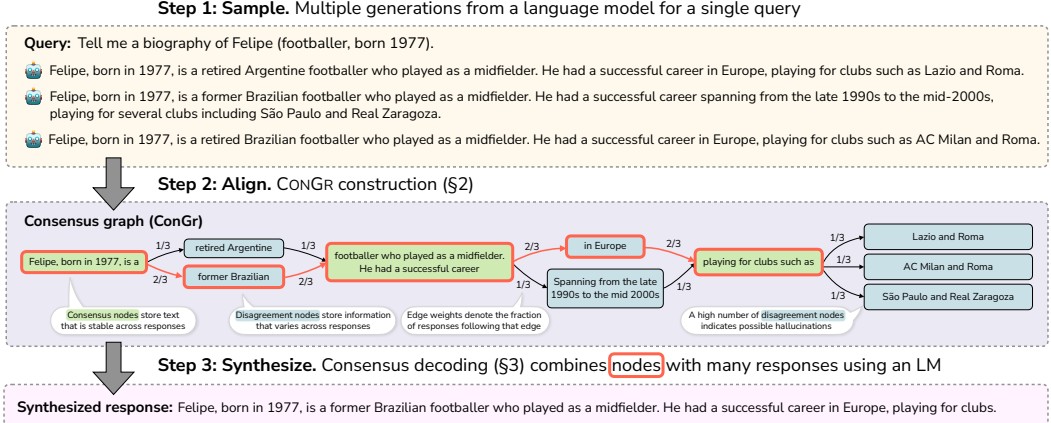

Figure 1: Consensus Graphs (CONGRS) capture the variation in a set of sampled LM responses. A CONGR is a weighted DAG of: consensus nodes for text spans present in all responses and disagreement nodes for lexical differences between responses. A node's weighted degree represents the fraction of responses which contain the information in that node. In the above example task of generating factual biographies from 3 sampled responses, disagreement nodes with lower weighted degree might indicate possible hallucinations. For this reason, none of the information in the disagreement nodes is included in the final synthesized response (further details in §3).

Recent findings suggest that post-training methods such as RLHF significantly reduce the textual diversity among sampled responses (Kirk et al., 2024; Lake et al., 2024; West & Potts, 2025). This results in *anchor spans*: sequences of words that occur in the same order across response samples to the same prompt, "serving as textual scaffolding" (Li et al., 2024b). Post-trained model responses for the tasks we consider in this paper exhibit this structural lexical overlap, too. To demonstrate this, we sample five responses each from 50 examples from a biography generation task from an aligned model (Qwen2.5-72B-Instruct) and split each response into ordered segment quantiles. For each segment position, we measure the lexical similarity across the set of response segments at that position. Figure 2 shows that the similarity between different ordered segments of the responses is higher than a control setting where the generations are randomly shuffled at a sentence

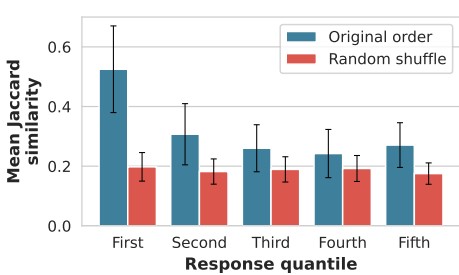

Figure 2: Responses for a biography generation task from an aligned model (Qwen2.5-72B-Instruct) contain lexical overlap across ordered segments of responses compared to a shuffled baseline. Differences in lexical similarity (measured with Jaccard similarity over word sets) are significant as measured using a paired $t$-test for all quantiles.

level. Could these anchor spans correspond to the epistemic signals that synthesize information?

Building on this observation, we introduce Consensus Graphs (CONGRS), a data structure to map the variation across a set of LM responses, identifying regions of consistent and inconsistent testimony (Figure 1). Specifically, we construct CONGRS as a directed acyclic graph of response text, where each path through the graph corresponds to a single response. We adapt the Needleman-Wunsch algorithm (Needleman & Wunsch, 1970; Lee et al., 2002), a foundational Multiple Sequence Alignment (MSA) algorithm from bioinformatics (Edgar & Batzoglou, 2006), to efficiently identify anchor spans shared by all responses (consensus nodes in Figure 1), followed by surface-level analysis by an LM judge.

For synthesizing a single final response from a CONGRS data structure, we present two complementary strategies, each suited to different epistemic contexts (§3). *Aggregation algorithms* take a consensus-focused approach to response synthesis and are designed for contexts such as biography generation where individual claims are often independent and may not logically depend on others. We design *consensus decoding*, an aggregation algorithm to synthesize novel responses that

reduce hallucinations while retaining complementary information distributed across responses. *Intervention algorithms* take a disagreement-focused approach to response synthesis by using sampled responses as "drafts" and then using extra analysis on regions of high disagreement as indicated by CONGRS to synthesize a new response. These are appropriate for contexts such as mathematical or logical reasoning, where maintaining global coherence is important and where variation can point to regions in responses requiring further targeted analysis. We introduce *guided self-verification*, an intervention algorithm to increase performance on reasoning tasks by aiding the challenging task of LM self-verification (Huang et al., 2024b) through improving error localization.

We present experiments on three kinds of tasks where high variation across responses provides rich epistemic signals: long-form biography factuality, refusal tasks that require abstaining from unanswerable queries, and mathematical reasoning. For the task of generating factual biographies, consensus decoding improves factual precision by up to 56% over an average response. It also achieves comparable performance to an alternate method that heavily uses LMs for claim analysis (Jiang et al., 2024) at only 20% of the cost in secondary LM API calls. For refusal-based tasks of unanswerable queries, consensus decoding substantially increases abstention rates, with up to a 56% increase in abstention and a 58% decrease in hallucination rate. For reasoning tasks, guided self-verification uses the variability captured by CONGRS to localize errors, boosting performance on MATH by up to 6 points of accuracy beyond self-consistency (Wang et al., 2023b) and self-verification (Zhao et al., 2025; Tyen et al., 2024; Weng et al., 2023). Overall, our results show how CONGRS can mitigate the absence of traditional metadata to establish information reliability by capturing the epistemic signals provided by LM response variation, allowing efficient response synthesis that out-performs any singular LM response, without requiring a reasoning model.

## 2 Capturing LM Response Variation with Consensus Graphs

We introduce Consensus Graphs (CONGRS), a data structure to capture the lexical and semantic variation across a set of sampled LM responses to the same prompt. To construct CONGRS, we first use purely lexical alignment to identify the *anchor spans* of shared text across all responses. We then use a secondary LM as a judge to determine which parts of the responses *between* the anchor spans are semantically equivalent. Lexical alignment limits our reliance on secondary LMs, greatly reducing cost over alternative methods (see §4.1) and limiting additional hallucinations introduced by secondary LMs. The resulting CONGR represents the variability of the original responses in a way that permits efficient synthesis of the information contained in different candidate responses.

**Definition.** Given a set $R$ with $|R| = m$ responses sampled from an LM $\mathcal{M}$ given a prompt $x$, a consensus graph (CONGR) is a weighted directed acyclic graph $G(R) = (V, E)$. Each response $r \in R$ corresponds to a path through $G(R)$. The set of nodes, $V = V_C \cup V_D$ includes both consensus nodes, $V_C$, and disagreement nodes, $V_D$. Each node $v \in V$ has corresponding text $v_{\text{text}}$. Consensus nodes $v \in V_C$ contain anchor spans present across all $|R|$ responses (green nodes in Figure 1). Disagreement nodes $v \in V_D$ contain information that varies across responses (blue nodes in Figure 1). Each edge $e \in E$ has a weight in $(0, 1]$ corresponding to the fraction of responses that follow that edge.

**Step 1: Generating lexical partial order graphs.** The first step in constructing a CONGR is finding anchor spans, common lexical subsequences among the set of model responses $R$. To do so, we adapt the Needleman-Wunsch (NW) algorithm (Needleman & Wunsch, 1970), a dynamic programming algorithm originally developed to align biological sequence data. While Li et al. (2024b) also use MSA to perform sequence alignments, we perform several modifications to the standard versions of these algorithms. In particular, we use the approach of Lee et al. (2002) to create a lexical DAG from the alignments produced by running NW. The algorithm has the objective of minimizing a misalignment penalty between responses, finding the optimal way to match tokens regardless of variations in relative positioning and ordering across responses. We adapt NW to process lexical token sequences[1] as opposed to the single characters of protein sequences, and we also add a gap penalty to the dynamic programming objective. This algorithm takes under a second to run across all of our experimental settings, and has a time complexity of $O(mL^2)$, where $m$ is the number of responses and $L$ is the length of the longest response. Further details are in Appendix B.

---

[1] In this context, *tokens* are full words for ease of decoding responses.

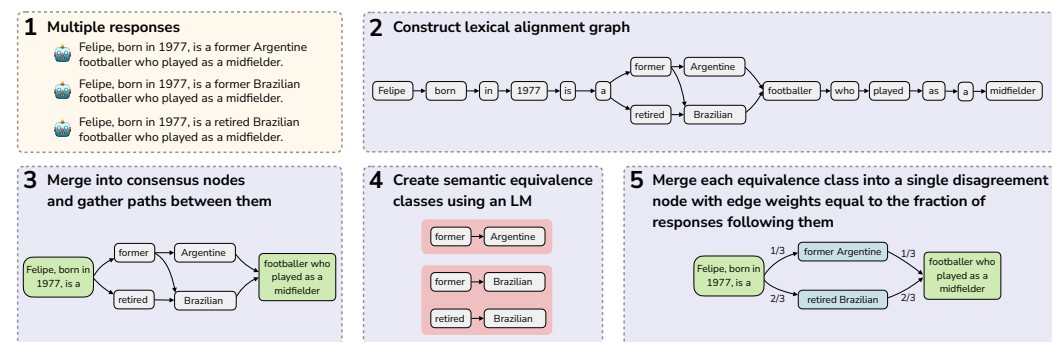

Figure 3: From a set of responses, we construct a CONGR by 1) Using Needleman-Wunsch (Lee et al., 2002) to construct a lexical DAG where each node's text is a single token, 2) Merging consecutive sequences of nodes that are present in all responses to create consensus nodes, 3) Extracting paths between consecutive pairs of consecutive nodes and using a LM to create semantic equivalence classes 4) Creating a disagreement node for each semantic equivalence class.

This step results in a weighted lexical DAG, with a node's weighted degree corresponding to the proportion of sequences containing that node. As seen Panel 2 of Figure 3, there is no distinction yet between consensus and disagreement nodes, and each node contains only one token. Nonetheless, this structure already begins to show where responses converge and diverge.

**Step 2: Creating consensus nodes.** To create consensus nodes, we merge sequences of nodes that are present in all generations. For each such sequence $s = v_1, v_2, ..., v_k$, we: 1) create a consensus node $c$ with $c_{\text{text}}$ equal to the concatenation of all tokens in the sequence, 2) replace each sequence with its corresponding consensus node, and 3) preserve edge connections at the beginning and end of each sequence. All edges that entered $v_1$ now enter $c$, and all edges that exited $v_k$ now exit $c$. Panel 3 of Figure 3 shows how the graph appears at the end of this step.

**Step 3: Creating disagreement nodes.** Now that the graph contains consensus nodes, we consider the multiple directed paths between each pair of consecutive consensus nodes $(c_i, c_{i+1})$ which represent how the original responses diverge between anchor spans. The final step in creating a CONGR is determining whether the text corresponding to paths between consensus nodes are semantically equivalent. We use a secondary LM to perform pairwise comparisons between each path between consecutive consensus nodes $(c_i, c_{i+1})$ to create semantic equivalence classes, which are groups of paths judged to convey the same meaning despite different wording, as in Panel 4 of Figure 3. For each equivalence class, we create a disagreement node $v \in V_d$ with the path with the longest text chosen for $v_{\text{text}}$. We preserve alternative phrasings as additional metadata in the node, in order to be able to reconstruct the original responses as required by decoding methods described in §3. We then replace nodes that are in between each pair of consecutive consensus nodes with the newly created disagreement nodes, as shown in Panel 5 of Figure 3. This process results in an alternating structure of consensus and disagreement nodes. Appendix C gives details such as prompts, judgment criteria, and pseudocode for this CONGR construction step.

Table 1: Statistics for CONGRS constructed from five Qwen2.5-72B-Instruct responses per evaluation example, averaged across nodes in each CONGR and averaged across examples. C: consensus nodes, D: disagreement nodes.

| Mean statistics→ | # Nodes | %C | %D | # Words C | # Words D | # Branches C |
|---|---|---|---|---|---|---|
| Biographies | 27.50 | 27 | 73 | 7.70 | 25.71 | 2.39 |
| Long-form PopQA | 25.41 | 27 | 73 | 5.38 | 20.12 | 2.11 |
| False Presuppositions | 15.84 | 11 | 89 | 2.40 | 2.22 | 0.55 |
| Scientific References | 93.22 | 3 | 97 | 3.09 | 25.83 | 1.70 |
| Historical Events | 21.27 | 35 | 65 | 14.29 | 17.95 | 1.95 |
| AIME | 133.37 | 25 | 75 | 3.68 | 38.30 | 3.21 |
| MATH | 47.22 | 31 | 69 | 2.67 | 36.06 | 2.29 |
| Creative Writing | 43.40 | 30 | 70 | 2.26 | 25.27 | 2.47 |

**Descriptive statistics of CONGRS.** We find that there is in fact enough shared structure in aligned model responses to be sufficiently captured by CONGRS (Table 1). For the seven datasets we con-

sider (details in §4), we find that responses share enough text spans to construct consensus nodes. For example, a CONGR for the biography generation task has, on average, 7 anchor spans (which are captured by consensus nodes) consisting of 7.7 words each. For the same task, 28.0% of consensus nodes contain only stopwords, and only 1.8% of disagreement nodes contain only stopwords. Only 6 out of 100 examples yield responses with so much variability that no consensus nodes are made. For the mathematical reasoning tasks, MATH and AIME, there are no degenerate graphs containing zero consensus nodes. While reasoning solutions can diverge, there is consensus in the form of explicit mentions of solution steps and common intermediate calculations. We also created CON-GRS for a set of 100 creative writing prompts from WritingPrompts (Fan et al., 2018). Even in this highly open-ended setting, we can successfully construct CONGRS, with the caveat of consensus nodes containing much less text. However, we note that for this setting, response variation does not necessarily carry epistemic signals such as competing factual claims or diverging reasoning paths. Thus, while there is still evidence of the existence of anchor spans in aligned model responses for open-ended tasks, CONGRS may offer more limited utility in that setting.

## 3    SYNTHESIZING RESPONSES FROM CONSENSUS GRAPHS

We show that CONGRS can be used for generating final responses by synthesizing relevant information across a set of sampled LM responses. We design two complementary algorithms that synthesize novel responses using CONGRS, each suited towards different contexts. *Consensus Decoding* is an *aggregation algorithm* that takes a consensus-focused approach to response synthesis by using CONGRS to select high-degree nodes containing information present in multiple responses. Aggregation algorithms like consensus decoding are appropriate for contexts

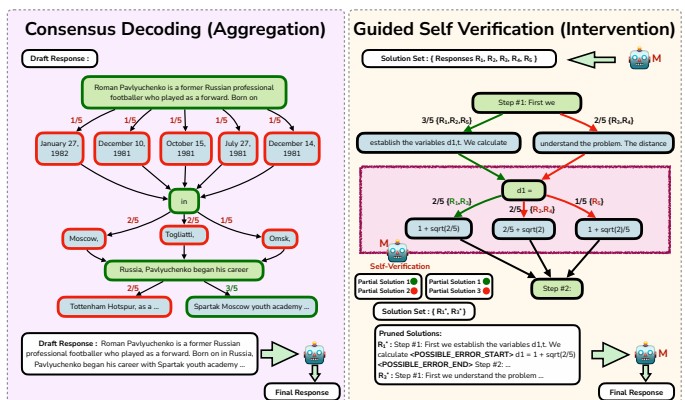

Figure 4: Consensus decoding (left) uses a CONGR to combine text present in many responses. Guided self-verification (right) uses a CONGR to localize possible errors in reasoning chains.

where constituent parts of a response can be independently decomposed and recombined. We also design an algorithm called *guided self-verification*. Guided self-verification is an *intervention algorithm* that takes a disagreement-focused approach by treating full responses as "drafts" and using CONGRS to identify regions of high disagreement between responses. This type of algorithm can then leverage additional computation to guide the model to resolve disagreements and modify the set of draft responses. They are suited to contexts where the relationship between different parts of responses needs to be preserved. Pseudo-code and prompts for each algorithm are in Appendix E.

**Consensus decoding.** This algorithm generates a synthesized response from a CONGR by incorporating text present in many of the original responses We start with a set $R$ with $|R| = m$ responses and a hyperparameter consensus threshold $\tau \in [0, 1]$, where higher values mean that a text span must be present in more responses to make it into the final response. Consensus decoding traverses the CONGR in topological order and selects nodes with weighted degrees of at least $\tau$, i.e., including text present in at least $\tau m$ of the original responses. Then, the text in each selected node is concatenated to form a draft response. Since this can result in disfluencies, we prompt the same secondary LM that was used for CONGR construction to fix grammatical errors. This step also includes explicit instructions to abstain when the selected content is too fragmented to produce a coherent response. In practice, we find via manual inspection that this final task is constrained enough that the secondary LM does not introduce or remove hallucinations. Across a sample of 100 biography examples, a qualitative analysis shows that only five claims are dropped from a final response, two claims are modified, and two claims are added. (details in Appendix F). Figure 4 (left) shows an example of consensus decoding.

**Guided self-verification.** Maintaining step-by-step coherence in reasoning chains is crucial, and therefore, consensus decoding may not be suitable. Self-verification—directly prompting a model to assess which of its own responses are correct—is a simple form of test-time scaling (Zhao et al., 2025). However, LMs often struggle at self-verification because they struggle to *localize errors* in their own generations (Kamoi et al., 2024; Tyen et al., 2024; Zhang et al., 2024; Huang et al., 2024a). Our guided self-verification method uses a CONGR to help a model localize its own possible errors.

Given a set $R$ with $|R| = m$ responses generated by model $\mathcal{M}$ and a pruning threshold hyperparameter $\kappa \in [0, 1]$, we initialize a set of candidate responses $\mathcal{C}$ with $R$ and traverse the CONGR in a topological order, marking consensus nodes which are followed by at least $\kappa m$ disagreement nodes. These regions of high variability may correspond to different valid reasoning steps or to an error. To distinguish between these cases, we provide $\mathcal{M}$ with each partial solution up to the next consensus node and prompt it to self-verify any error. Erroneous partial responses are pruned from $\mathcal{C}$. After repeating this process at all uncertain nodes, we use the remaining responses in $\mathcal{C}$ as new context to prompt $\mathcal{M}$ to synthesize a final response (e.g., the final answer to a reasoning problem). Figure 4 (right) illustrates this approach.

# 4 Synthesizing with ConGrs Improves Downstream Performance

We demonstrate how decoding from CONGRS can be applied to three categories of tasks: improving the factuality of long-form responses about named entities ( §4.1), abstain on unanswerable queries, even when the original set of responses contains no abstentions (§4.2), and improve performance on reasoning tasks ( §4.3).

## 4.1 Consensus Decoding Improves Factuality of Long-Form Generations

**Models sampled from for aggregation.** We generate $m = 5$ responses each from six RLHF-aligned models across three model families and two sizes: Llama-3.3-70B-Instruct (Grattafiori et al., 2024), Qwen2.5-72B-Instruct (Yang et al., 2024), OLMo-2-32B-Instruct (OLMo et al., 2024), Llama-3.1-8B-Instruct, Qwen2.5-7B-Instruct, and OLMo-2-7B-Instruct. We use a temperature of 0.9. The secondary LM for CONGR construction is GPT-4.1-Mini. Appendix G contains detailed model configurations, prompts, and details about runtime and computational resources. We present results for the three larger models in the main paper and three smaller models in Appendix H.

**Methods.** We compare to: mean scores of the original responses, *Greedy decoding*, *MBR Decoding* (Bertsch et al., 2023), *Shortest response* (Dimakis, 2025), *Reasoning model response* from QwQ-32B (Yang et al., 2024), and an LM-generated consensus from the responses without a CONGR. We also compare to *Atomic Self-Consistency* (ASC, Thirukovalluru et al. 2024), a synthesis method that uses sentence clustering. Detailed method descriptions and prompts are in Appendices G and H. We report mean and standard deviation over five replications.

**Datasets and evaluation.** We apply consensus decoding to two benchmarks: 1) biography generation for entities from FActScore (Min et al., 2023) and 2) long-form PopQA (Mallen et al., 2023). Like Jiang et al. (2024), we transform PopQA into a long-form task by prompting models to generate a paragraph of facts for each entity. For each dataset, we randomly sample 100 entities. We use the FActScore metric for evaluation, which measures the fraction of facts in a response deemed to be supported in a reference text by an LM judge (Min et al., 2023). We report the number of supported and unsupported facts per response, averaged across entities. Both statistics are important since it is trivial to achieve a high FActScore by outputting no facts or a low number of facts. We also report the response ratio, i.e., the fraction of responses without an abstention. We summarize the factual precision (FActScore) and factual recall (number of supported facts) of methods with response ratios using the same approach as Jiang et al. (2024): if a method abstains for an entity, then that entity gets a FActScore of 1 and a number of supported facts of 0.

**Results.** Figure 5 shows that consensus decoding with CONGRS consistently yields a better trade-off between FActScore and number of supported facts than baselines for both datasets. At nearly all settings of the selection threshold $\tau$, consensus decoding is at the Pareto frontier of factual precision and factual recall. Full numerical results are in Table 8 and Table 9 (Appendix H). Appendix D

performs a qualitative analysis of CONGR construction in this setting, demonstrating that consensus nodes cover a variety of syntactic units and lengths and are more likely to contain supported facts.

Our method achieves higher performance than the LM consensus baseline, which only sometimes shows improvements over other methods but struggles to consistently improve factuality. This shows that explicitly modeling variability with CONGRS is more useful than simply having another LM summarize the consensus. Greedy and shortest responses are also not necessarily more factual than the base responses (which were generated using a temperature of 0.9), demonstrating the inconsistent utility of simple heuristic approaches to improve factuality. In addition, while QwQ-32B produces a high number of supported facts on average, it also produces a high number of unsupported facts. This indicates the limitations of its extended reasoning for factuality. Consensus decoding has higher FActScores while providing more supported facts than the corresponding ASC setting (e.g., $\tau = 0.3$ corresponds to $\Theta = 2$), showing that modeling variability at a more granular resolution than the sentence level improves performance. Furthermore, consensus decoding outperforms guided self-verification in this setting, showing that combining reliable information across multiple responses is more effective than refining a single response in this setting. For example, consensus decoding with $\tau = 0.3$ on Qwen2.5-72B-Instruct responses yields both a higher FActScore (0.72 vs. 0.71) and many more supported facts (20.8 vs. 15.3) than guided self-verification.

Choosing a selection threshold $\tau$ is a matter for the user to decide whether their use-case requires a) higher factuality and fewer facts or b) more facts at the expense of lower factuality. When consensus decoding abstains, the original responses had low factuality. For example, responses from Qwen2.5-72B-Instruct on average have a FActScore of 0.68, but the average FActScore of responses that lead consensus decoding (with $\tau = 0.3$) to abstain have a mean FActScore of only 0.31 (details in Appendix H, Table 7). Additional analysis regarding the effects of $\tau$ is in Appendix I and explicitly demonstrates how $\tau$ controls the tradeoff between factuality and informativeness in synthesized responses. In particular, our experiments show that using a lower value of $\tau$ is beneficial for very common entities, where responses are more likely to contain true but complementary facts. On the other hand, for rare entities, higher values of $\tau$ are preferred to ensure reliability. We provide ablations for the temperature used to generate $R$ as well as the number of responses $|R|$ in Appendix I. Results demonstrate that our method is effective even at very high temperatures (greater than 1) and using a higher number of original responses. We also provide an ablation using a small, open model as the external LM in Appendix I, demonstrating that our method is robust to choice of external LM for CONGR construction and the consensus decoding edit step.

Consensus decoding also outperforms beam search and the following variants: Diverse Beam Search (Vijayakumar et al., 2018) and Range Voting (Borgeaud & Emerson, 2020), for Qwen2.5-72B on biography generation; see results in Table 14 (Appendix H). This demonstrates the utility of performing post-hoc synthesis using full sampled responses.

**CONGRS are cost-effective.** Consensus decoding uses fewer tokens from secondary LMs than an alternative. Uncertainty-Aware Decoding (UAD) with closeness centrality is a successful approach to response synthesis for factuality that uses a secondary LM to split responses into claims (Jiang et al., 2024). Due to the cost of running UAD, we compare our method using a smaller

Table 2: Consensus decoding with CONGRS uses 82% fewer tokens from secondary LMs compared to Uncertainty-Aware Decoding (UAD) (Jiang et al., 2024) without sacrificing FActScore or response ratio.

| Method | FActScore | #T | #F | RR | Mean # Tokens |
|---|---|---|---|---|---|
| CONGRS | 0.62 | 19.00 | 8.63 | 0.96 | 13,391.88 |
| UAD | 0.59 | 27.32 | 20.84 | 0.96 | 76,220.48 |

evaluation set of 25 random entities. Table 2 shows that building CONGRS and running consensus decoding uses less than 20% as many secondary LM API tokens (13,392 vs. 76,220 per entity on average) while achieving a slightly higher FActScore and the same response ratio.

## 4.2 CONSENSUS DECODING ABSTAINS WHEN NO FACTUAL RESPONSE EXISTS

LMs can often produce confident yet hallucinated responses to queries that are unanswerable, such as queries that contain false presuppositions. While alignment can help address this issue, aligned LMs cannot currently always correctly abstain on these types of queries. We demonstrate how consensus

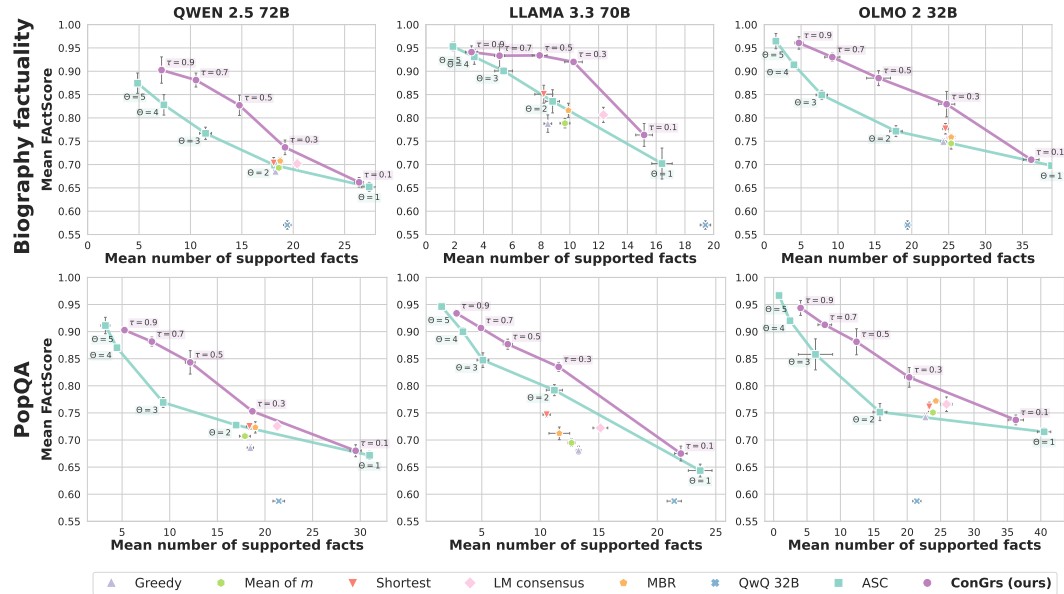

Figure 5: Consensus decoding with CONGRS achieves the best trade-off between FActScore (the fraction of a response's claims that are true) and the number of true claims provided. Up and to the right is better. $\tau$ is the selection threshold for consensus decoding. $\Theta$ is the analogous parameter for the ASC baseline. Top row: Biography factuality. Bottom row: PopQA.

Table 3: Consensus decoding with CONGRS consistently achieves lower Response Ratio and Hallucination Score than other response synthesis methods on the False Presuppositions and Scientific References HALoGEN tasks. R: Response Ratio, H: Hallucination Score. Lower is better for both.

| | False Presuppositions | | | | | | Scientific References | | | | | |
| | Qwen2.5-72B | | Llama-3.3-70B | | OLMo-2-32B | | Qwen2.5-72B | | Llama-3.3-70B | | OLMo-2-32B | |
| Method | R↓ | H↓ | R↓ | H↓ | R↓ | H↓ | R↓ | H↓ | R↓ | H↓ | R↓ | H↓ |
|---|---|---|---|---|---|---|---|---|---|---|---|---|
| Greedy | 0.22 | 0.16 | 0.66 | 0.49 | 0.00 | 0.00 | 0.78 | 0.67 | 0.32 | 0.30 | 0.50 | 0.41 |
| Mean of $m$ | 0.25 | 0.17 | 0.62 | 0.46 | 0.01 | 0.01 | 0.76 | 0.66 | 0.28 | 0.26 | 0.61 | 0.54 |
| Shortest | 0.16 | 0.11 | 0.68 | 0.49 | 0.01 | 0.00 | 0.61 | 0.55 | 0.16 | 0.16 | 0.30 | 0.26 |
| LM Consensus | 0.33 | 0.23 | 0.70 | 0.51 | 0.02 | 0.01 | 0.33 | 0.29 | 0.29 | 0.26 | 0.08 | 0.07 |
| MBR | 0.22 | 0.15 | 0.62 | 0.45 | 0.00 | 0.00 | 0.75 | 0.66 | 0.24 | 0.22 | 0.65 | 0.58 |
| QwQ-32B | 0.04 | 0.01 | 0.04 | 0.01 | 0.04 | 0.01 | 0.83 | 0.74 | 0.83 | 0.74 | 0.83 | 0.74 |
| CONGRS ($\tau$=0.3) | 0.14 | 0.09 | 0.33 | 0.21 | 0.00 | 0.00 | 0.19 | 0.16 | 0.16 | 0.15 | 0.05 | 0.04 |
| CONGRS ($\tau$=0.5) | 0.11 | 0.07 | 0.27 | 0.17 | 0.00 | 0.00 | 0.11 | 0.08 | 0.08 | 0.07 | 0.02 | 0.02 |

decoding can mitigate this by identifying the high level of inconsistency among responses to these types of queries. In this section, we use the same models and baselines as in §4.1.

**Datasets and evaluation.** We evaluate consensus decoding on the abstention-based tasks from the HALoGEN benchmark (Ravichander et al., 2025). The False Presuppositions task evaluates whether LMs correctly refuse to answer when posed with a question that does not have a correct answer. The Scientific Attribution task evaluates whether LMs will hallucinate references in support of incorrect facts. The Historical References task evaluates whether LMs will hallucinate details of a fictional meeting between two historical figures who have never met. We report the same metrics used in Ravichander et al. (2025): Response Ratio, which measures how often a model responds to the initial prompt, and Hallucination Score, which measures hallucination rate.[2] We sample 250 instances for each task. Results for Historical Events are located in Appendix H and show similar trends.

---

[2]Ravichander et al. (2025) also report a *Utility Score*, but it is always equal to 1 minus Response Ratio here.

**Results.** Table 3 shows that consensus decoding with CONGRS consistently reduces both Response Ratio and Hallucination Score compared to baselines for False Presuppositions. For Qwen2.5-72B-Instruct, consensus decoding with a selection threshold of $\tau = 0.5$ reduces Response Ratio by 56% (from 0.25 to 0.11) and Hallucination Score by 58% (from 0.17 to 0.07). Consensus decoding both improves abstention and results in more reliable responses when not abstaining. These improvements hold across model size. QwQ-32B is a strong baseline for this task, indicating the utility of its extended reasoning for queries with false presuppositions. For Scientific References, consensus decoding reduces Response Ratio and Hallucination Score as well. For Qwen2.5-72B-Instruct, there is an 86% improvement in Response Ratio (from 0.76 to 0.19) and an 88% improvement in Hallucination Score (from 0.66 to 0.08). QwQ-32B produces many hallucinated references, even after extended reflection.

### 4.3 GUIDED SELF-VERIFICATION IMPROVES PERFORMANCE ON REASONING TASKS

**Experimental setup.** We evaluate our guided self-verification method using all questions from the test split of Berkeley MATH (Hendrycks et al., 2021) and all questions from AIME 2024 (MAA, 2024). We report accuracy. We generate responses only from Llama-3.3-70B-Instruct and Qwen2.5-72B-Instruct due to extremely low baseline performance of the smaller models in our collection. Appendix G contains model configurations and specific prompts. We compare guided self-verification with CON-GRS against the following methods:*Self-consistency* (Wang et al., 2023b), *Self-verification* (Zhao et al., 2025; Tyen et al., 2024; Weng et al., 2023). These methods are intended to achieve perfor-

| Method | Qwen2.5-72B | | Llama-3.3-70B | |
|---|---|---|---|---|
| | MATH | AIME | MATH | AIME |
| Self-Consistency | 0.68 | 0.20 | 0.59 | 0.23 |
| Self-Verification | 0.66 | 0.15 | 0.56 | 0.19 |
| Pairwise Self-Verification | 0.69 | 0.15 | 0.59 | 0.24 |
| CD w/ CONGRS ($\tau$=0.5) | 0.68 | 0.20 | 0.61 | 0.23 |
| GSV w/ CONGRS ($\kappa$=0.7) | 0.70 | 0.20 | 0.65 | 0.27 |
| Pass@$m$ | 0.74 | 0.20 | 0.70 | 0.33 |
| QwQ-32B | 0.91 | 0.50 | 0.91 | 0.50 |

Table 4: Guided self-verification (GSV) with CONGRS consistently outperforms Self-Consistency and Self-Verification. Pass@$m$ is an upper bound, and so is the response from a more expensive reasoning model, specifically trained to excel at similar tasks, QwQ-32B.

mance parity with Pass@$m$ without the use of trained external verifier models. We also include results from QwQ-32B, a dedicated serial reasoning model that also provides an upper bound.[3] More detailed descriptions of these methods are contained in Appendix G. For all methods, we aggregate across the same number of responses ($m = 5$) generated with a temperature of 0.9.

**Results.** Results in Table 4 show that for MATH (Hendrycks et al., 2021), guided self-verification achieves 2 and 6 point accuracy gains over self-consistency for Qwen2.5-72B-Instruct and Llama-3.3-70B-Instruct respectively, as well as larger gains against vanilla self-verification. On the much more challenging AIME (MAA, 2024) dataset, guided self-verification achieves a 4 point gain over self-consistency for Llama-3.3-70B-Instruct. Guided self-verification narrows the gap between self-consistency and Pass@$m$, which demonstrates that even at small values of $m$, a comparative analysis of diverse reasoning chains can help extract additional information to improve performance. Notably, guided self-verification improves upon vanilla self-verification and pairwise self-verification without CONGRS (Zhao et al., 2025). It also surpasses consensus decoding in this setting. Localizing possible errors with CONGRS aids strong models in identifying the potential errors that they made, in the absence of trained reward models or models such as QwQ-32B that are explicitly trained for reasoning and can generate many redundant tokens (Chen et al., 2024b).

## 5 CONCLUSION

We introduce CONGRS, a DAG-based data structure that represents the variability in a set of LM responses, sampled for any single prompt. By combining fast, purely lexical sequence alignment with minimal use of secondary LMs, CONGRS are constructed efficiently. We demonstrate how the structure of CONGRS allows for simple LM response synthesis that aggregates epistemic signals

---

[3]Results are sourced from Qwen Team (2024).

across responses, in the absence of external metadata. We also show how CONGRS can be used to guide self-verification for reasoning problems. We hope future work will explore how variation can also encode alternate perspectives and creative possibilities for more open-ended tasks, as well as for responses from different models. Our findings highlight how CONGRS can be a flexible, efficient, yet simple approach to study and exploit LM response variation, as a step towards building more reliable LM systems.

## REPRODUCIBILITY STATEMENT

The supplementary materials include source code for constructing CONGRS and reproducing our experiments. Additionally, all algorithm details for CONGRS construction are in Appendices B and C. Appendix E contains details for consensus decoding and guided-self verification. Appendix G contains hyperparameters and prompts for all the baselines we compare to.

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

## A   ADDITIONAL RELATED WORK

Our general approach aligns with prior work on collective intelligence (Hong & Page, 2004), the "wisdom of the crowd" effect (Kremer et al., 2014; Suzgun et al., 2023), and ensembling in machine learning (Zhou et al., 2002), which demonstrates that multiple diverse perspectives often outperform any single one.

**Inference-time scaling.**   Inference-time scaling strategies vary widely, with most selecting one response and only operating on short responses. Selection methods include Minimum Bayes Risk (MBR) decoding (Bertsch et al., 2023; Suzgun et al., 2023), Universal Self-Consistency (Chen et al., 2024c), and rejection sampling/Best-of-N sampling (Liu et al., 2024; Brown et al., 2025; Snell et al., 2025). Other methods aggregate one model's responses using sophisticated intermediate generation structures (Yao et al., 2023; Saha et al., 2024; Xie et al., 2023) that decode partial sequences in parallel. Xu et al. (2022) uses lattices to generate diverse responses during decoding and relies on hypothesis recombination in a manner similar to steps in CONGR construction, except within the context of a search procedure. Some approaches also aggregate across responses from multiple LMs (Wang et al., 2025; Chen et al., 2024a) through multi-agent debates. In comparison, our methods sample full independent responses from one model and then synthesize novel responses.

**LM response factuality.**   Several methods improve long-form response factuality from LMs. Many involve additional fine-tuning (Tian et al., 2023), which has mixed results in terms of consistent improvement (Gekhman et al., 2024; Kang et al., 2025). Some methods involve only sampling one generation with novel decoding algorithms (Chuang et al., 2024; Lee et al., 2022). Others rely on discussions between multiple LLMs (Du et al., 2024; Cohen et al., 2023; Yoffe et al., 2024) or self-correction and re-ranking strategies (Dhuliawala et al., 2024; Wei et al., 2024; Li et al., 2024c;a; Manakul et al., 2023; Jain et al., 2024). Some methods create claim-level uncertainty estimates based on multiple sampled generations and use them to filter out low-confidence claims (Jiang et al., 2024; Manakul et al., 2023; Farquhar et al., 2024; Jiang et al., 2025; Mohri & Hashimoto, 2024; Thirukovalluru et al., 2024; Wang et al., 2024a;b). Compared to methods requiring claim/sentence level decompositions, CONGRS can be used to combine spans of variable lengths within responses.

**LM abstention.**   The ability of LMs to abstain from answering uncertain queries is a critical mechanism for reducing the effect of hallucinations and improving safety in high-risk applications (Tomani et al., 2024; Wen et al., 2024; Feng et al., 2024; Brahman et al., 2024). Existing methods for abstention, especially for long-form responses, are brittle. In particular, alignment-based approaches can cause over-refusal of queries (Brahman et al., 2024), while the effectiveness of prompt-based methods varies widely since it relies on the ability of models to self-reflect about their own uncertainty (Kadavath et al., 2022; Wang et al., 2023a). Methods that rely on uncertainty quantification metrics (either from the original LM itself or other methods) incur additional cost by requiring held-out sets for calibration (Jiang et al., 2024; Mohri & Hashimoto, 2024; Jiang et al., 2025; Farquhar et al., 2024). In contrast, consensus decoding (Section 3) allows for a simple way to perform abstention without any additional supervised data or calibration procedures.

## B   NEEDLEMAN-WUNSCH HYPERPARAMETERS

We apply the Needleman-Wunsch multiple sequence alignment algorithm from Lee et al. (2002). We create partial order graphs in an iterative fashion by aligning all sequences one by one. In our case; each node in the graph represents a token (full word). In order to align an incoming response to the intermediate partial order graph; we visit graph nodes in the order of the topological sort and consider all valid positions for insertion through a dynamic programming approach. In particular, we select an insertion which minimizes the total cost of aligning an incoming sequence to an intermediate lexical partial order graph. The cost of two aligned positions between the graph and the incoming sequence is determined by string similarity if both positions correspond to nodes; and by a gap penalty if one or both positions correspond to gaps. We use an affine gap penalty scheme as it is better suited for aligning our lexical sequences. We use the following hyper-parameters for our alignment: $gap\_open\_penalty = -1$, $gap\_extend\_penalty = -1$, $match\_penalty = 1$, $mismatch\_penalty = -2$. Hence, we adapt the implementation of Lee et al. (2002) by aligning lexical sequences instead of

protein sequences and introducing a gap penalty to the cost function. This procedure has time complexity $O(mL^2)$, where $m$ is the number of responses and $L$ is the maximum response length. In all settings of our experiments, the clock time to run this algorithm is under one second.

## C CONGRS CONSTRUCTION PROMPTS

We present the pseudocode for creating disagreement nodes in Algorithm 1.

---

**Algorithm 1:** Creating Disagreement Nodes

---

**Input:** Set of $m$ responses $R$, Graph $g(R) = (V, E, W)$ with consensus nodes
      $V_C = \{c_1, c_2, ..., c_k\}$
**Output:** Complete CONGR with disagreement nodes
$V_D \leftarrow \emptyset$
**for** *Each consecutive pair* $(c_i, c_{i+1})$ **do**
    $P_{i,i+1} \leftarrow \{p_1, p_2, ..., p_l\}$ ;            `// Paths between c_i and c_{i+1}`
    **for** *each path* $p_j \in P_{i,i+1}$ **do**
        $p_{j_\text{text}} \leftarrow$ ""
        **for** *each node* $v \in p_j$ **do**
            $p_{j_\text{text}} \leftarrow p_{j_\text{text}} \circ v_\text{text}$
        **end**
    **end**
    $\{E_1, E_2, ..., E_n\} \leftarrow \text{SemanticEquivalenceClasses}(P_{i,i+1})$ ;     `// Using LM`
    **for** *each equivalence class* $E_q \in \{E_1, E_2, ..., E_n\}$ **do**
        Create disagreement node $v_q$ with $v_{q_\text{text}} \leftarrow p_{j_\text{text}}$ for some $p_j \in E_q$;
        Store alternative phrasings $S(v_q) \leftarrow \{p_{j_\text{text}} | p_j \in E_q\} \setminus \{v_{q_\text{text}}\}$;
        Add edges $(c_i, v_q)$ and $(v_q, c_{i+1})$ to $E$, preserving original weights;
    **end**
    Add $v_q$ to $V_D$;
**end**
**return** *Complete* CONGR $g$

---

We present prompts to do a pairwise comparison to create equivalent classes with a secondary LM (GPT-4.1 mini) below:

---

**Comparison prompt for disagreement node construction for text**

You are given two pieces of text. Your task is to determine whether they are semantically equivalent based solely on their factual content.
Here are the specific guidelines:
- Texts are equivalent if they convey the same core information or concept, regardless of wording or structure
- If one text has information that is a subset of the other text, then the texts are equivalent
- Focus ONLY on the essential claims, not on:
  * Stylistic differences or tone
  * Level of detail (if the core facts remain the same)
  * Connotative differences between words
  * Implied significance or emphasis
  * Presentation order (if all key information is present in both)
- Minor additions of non-contradictory information should not make texts non-equivalent
- For ambiguous cases, prioritize the central claim or purpose of the text
Examples of equivalent pairs:
- 'The meeting starts at 3pm' and 'The 3 o'clock meeting will begin on time'
- 'Research indicates a 15% increase' and 'Studies show a fifteen percent rise'
- 'was influential in the field' and 'had a significant impact on the community'
Examples of non-equivalent pairs:
- "The project might be completed by Friday" and "The project will be finished by Friday"
- "Most experts agree on the approach" and "All experts support the approach"
Strictly follow these guidelines and return ONLY:
- equivalent
- not equivalent

---

**Comparison prompt for disagreement node construction for math**

You are given two pieces of text from mathematical solutions. Your task is to determine whether the two solution segments are mathematically equivalent in their content, while allowing for stylistic variations.

Here are some important guidelines:
- Solutions should be considered equivalent if:
  1. They communicate the same mathematical content/approach, even if word choice or phrasing differs
  2. They contain the same key mathematical ideas, even if expressed differently
  3. The same mathematical steps are described, even if using different words
  4. They present the same final answer, regardless of wording style or formatting
- Allow for these variations while still considering solutions equivalent:
  1. Stylistic differences ("we will" vs. "we'll" or "I'll")
  2. Different levels of formality in the explanation
  3. Minor rephrasing that preserves the core mathematical content
  4. Use of synonyms or alternative mathematical terminology for the same concept
- Solutions are NOT equivalent if:
  1. They use fundamentally different mathematical approaches
  2. They work with different formulas or equations
  3. They present different mathematical steps or operations
  4. They reach different conclusions or answers
  5. One contains substantial mathematical content that the other lacks
- When examining final answers, focus on mathematical equivalence rather than stylistic presentation
- For solution steps, maintain the core mathematical approach while allowing for rephrasing

Examples of solutions that SHOULD be considered equivalent:
- "We will systematically evaluate each possible grouping" and "We'll evaluate each grouping"
- "The answer is x = 5" and "Therefore, x equals 5"
- "Using the quadratic formula" and "Applying the quadratic formula"

Strictly follow the guidelines above.

Return your judgment in the following format. Do not include any other text:

  - equivalent

  - not equivalent

## D    CONGR QUALITATIVE ANALYSIS

In the box below, we present excerpts from an example set of biographies (generated using Qwen2.5-72B) for one of the entities in FActScore (Min et al., 2023). We construct a CONGR from this set of generations. The text in bold corresponds to claims contained in consensus nodes of the CONGR, and the non-bolded text corresponds to claims contained in disagreement nodes. The consensus nodes cover a variety of syntactic units and lengths, from single prepositions ("in") and simple noun-phrases ("Tottenham Hotspur") to complex noun and verb phrases ("Roman Pavlyuchenko is a former...").

For this particular example, all of the claims contained in consensus nodes is accurate[4]. On the other hand, most of the claims in disagreement notes is inaccurate. For instance, all five generations provide different birthdates. Indeed, none of them are accurate. On the other hand, claims contained in multiple (but not all) generations are more likely to be accurate. For instance, multiple generations claim that Pavlyuchenko transferred to Tottenham in 2008, which is accurate. On the other hand, while most of the generations contain the claim that he started his career in Spartak Moscow's youth academy, this claim is not accurate. However, CONGRS do capture the fact that this claim is less certain than claims contained in consensus nodes. As detailed in Section 3, the selection threshold hyperparameter $\tau$ can be used to control the desired level of consensus to create a synthesized generation.

---

**Example of a CONGR, bold font indicates consensus node text**

1. **Roman Pavlyuchenko is a former Russian professional footballer who played as a** forward. **Born on** January 27, 1982, **in** Moscow, **Russia, Pavlyuchenko began his career with** local club Spartak Moscow's youth academy and made his senior debut for the team in 2001. His performances caught the attention of European clubs, leading to **a high-profile move to Tottenham Hotspur.**

2. **Roman Pavlyuchenko is a former Russian professional footballer who played as a** striker. **Born on** October 15, 1981, **in** Togliatti, **Russia, Pavlyuchenko began his career with** Spartak Moscow's youth system and made his professional debut for the club in 2001. His performances attracted international attention, leading to a **high-profile move to** English club **Tottenham Hotspur**.

3. **Roman Pavlyuchenko is a former Russian professional footballer who played as a** striker. **Born on** December 10, 1981, **in** Togliatti, **Russia, Pavlyuchenko began his career with** Spartak Moscow's youth academy before making his professional debut for the club in 1999. In 2008, Pavlyuchenko made a **high-profile move to Tottenham Hotspur.**

4. **Roman Pavlyuchenko is a former Russian professional footballer who played as a** striker. **Born on** July 27, 1981, **in** Omsk, **Russia, Pavlyuchenko began his career with** local club FC Lokomotiv Omsk before moving to Spartak Moscow, where he became a prolific goalscorer and won the Russian Premier League title in 2004. His impressive performances at Spartak attracted international attention, leading to a **high-profile transfer to** English club **Tottenham Hotspur** in 2008.

5. **Roman Pavlyuchenko is a former Russian professional footballer who played as a** forward. **Born on** December 14, 1981, **in** Moscow, **Russia, Pavlyuchenko began his career with** Spartak Moscow's youth system and made his debut for the first team in 2001. He quickly established himself as a prolific goalscorer, helping Spartak win multiple domestic titles. His performances caught the attention of European clubs, **leading to a move to** English Premier League side **Tottenham Hotspur** in 2008.

---

[4]We determine factuality using the information contained in (Transfermarkt, 2025), a reputable database of information about footballers.

# E    DECODING ALGORITHM PROMPTS

Figure 6 gives examples for applying consensus decoding in two different scenarios. We present the pseudocode for our consensus decoding algorithm in Algorithm 4. We present the pseudocode for our guided self-verification algorithm in Algorithm 5. We use $d^w(v)$ to refer to a node's weighted degree and $d_{in}(v), d_{out}(v)$ to refer to unweighted in-degree and out-degree respectively, normalized by the number of responses $m$.

---

**Algorithm 2:** General Aggregation Framework

---

**Input:** CONGR $G(R) = (V, E)$, selection function $S$, global edit function $G$
**Output:** Synthesized response or abstention
Initialize selected_nodes $\leftarrow \emptyset$ **for** *each node $v$ in topological_order($V$)* **do**
    **if** $S(v) = include$ **then**
        | selected_nodes $\leftarrow$ selected_nodes $\cup \{v\}$
    **end**
**end**
draft_response $\leftarrow$ concatenate_text(selected_nodes) **return** $G(draft\_response)$

---

**Algorithm 3:** General Intervention Framework

---

**Input:** CONGR $g(R) = (V, E, W)$, intervention threshold $\kappa$, verification model $\mathcal{M}$
**Output:** Synthesized response or abstention
Initialize candidate_responses $\leftarrow R$
Initialize uncertain_regions $\leftarrow \emptyset$
**for** each consensus node $c$ followed by disagreement nodes $D$ **do**
    **if** $|D|/|$candidate_responses$| \geq \kappa$ **then**
        uncertain_regions $\leftarrow$ uncertain_regions $\cup \{(c, D)\}$
    **end if**
**end for**
**for** each $(c, $disagreement_set$)$ in uncertain_regions **do**
    candidate_responses $\leftarrow$ apply_intervention($c$, disagreement_set, candidate_responses, $\mathcal{M}$)
**end for**
**return** synthesize_from_candidates(candidate_responses, $\mathcal{M}$)

---

**Algorithm 4:** Consensus Decoding from CONGRS

**Input:** CONGR $G = (V, E)$ with
$V = V_C \cup V_V \cup \{v_{\text{START}}, v_{\text{END}}\}$,
Consensus Threshold $\tau \in [0, 1]$,
Edit Function $f$

**Output:** Consensus Response $y_{\text{consensus}}$

$y_{\text{consensus}} \leftarrow$ "";      // Initialize empty string

$V_{\text{ordered}} \leftarrow \text{TopologicalSort}(g)$ ;    // Order nodes

**for** *each node $v \in V_{\text{ordered}}$ in order* **do**
    **if** $d^w(v) \geq \tau$ **then**
        $y_{\text{consensus}} \leftarrow y_{\text{consensus}} \circ v_{\text{text}}$ ;
           // Concatenate text
    **end**
**end**

$y_{\text{consensus}} \leftarrow f(y_{\text{consensus}})$ ;      // Apply edits

**return** $y_{consensus}$

---

**Algorithm 5:** Guided Self-Verification using CONGRS

**Input:** CONGR $G(R) = (V, E)$ with
$V = V_C \cup V_V \cup \{v_{\text{START}}, v_{\text{END}}\}$ for
a response set $R$ of size $m$, LM $\mathcal{M}$
which generated $R$, Pruning
Threshold $\kappa \in [0, 1]$, Candidate
Solution set $\mathcal{C} \leftarrow R$

**Output:** Decoded Response

$V \leftarrow \text{TopologicalSort}(g)$ ;    // Order all nodes

**for** *each consensus node $u \in V_C$* **do**
    **if** $d_{out}(u) \geq \kappa$ **then**
        **for** *each following variable nodes*
        *$v_a$ and $v_b$ with their corresponding*
        *candidate solutions $C_a$ and $C_b$,*
        *such that $(u, v_a), (u, v_b) \in E$* **do**
           Prune $C_a$, $C_b$ from $\mathcal{C}$ based on
           verification scores from
           $\mathcal{M}(partial(C_a, v_a), partial(C_b, v_b))$
        **end**
    **end**
**end**

Synthesize final decoded response:
  $y_{\text{decoded}} = \mathcal{M}(\mathcal{C})$

**return** $y_{decoded}$

---

The prompt used to remove disfluencies and synthesize the final response from the draft response using a secondary LM (GPT-4.1 mini) is provided below:

> **Consensus Decoding: Final synthesis prompt**
>
> You are given a piece of text that is a part of a {task}. This text may contain some minor errors that make it incoherent as well as potentially redundant information. Your task is to fix the errors and make the text coherent. Then, remove any redundant information. Text: {text}
> If this is not possible because the text is just a fragment of a sentence, return "Abstain". If the text already claims a lack of knowledge about the topic, return "Abstain". Only return the cleaned up text. Do not include any other text:

The prompt for pairwise self-verification of partial solutions with the same LM $\mathcal{M}$ is given below:

---

**Guided Self-Verification: Pairwise self-verification prompt**

You will be given a problem and 2 partial solutions. Your task is to use comparison as an EFFICIENCY TOOL to quickly identify potential errors. You will be given guidelines to follow, and you will be penalized if you do not follow them.

Problem: {problem}

Partial Solution 1: {partial_solution_1} Partial Solution 2: {partial_solution_2}

- CRITICAL GUIDELINES:
    * DO NOT penalize a solution for being incomplete or having missing steps
    * DO NOT make a comparison of which solution is better
    * DO NOT consider steps incorrect just because they differ between solutions
    * DO NOT prematurely evaluate based on final answers or future steps
    * DO NOT expect both solutions to be at the same stage of completion
    * DO NOT consider a step incorrect just because it lacks sufficient detail or justification
- KEY EFFICIENCY PRINCIPLE:
    * Use agreement between solutions as evidence of correctness
    * Use disagreement as a signal to investigate more deeply
    * Only label a step as an error if it contains a specific mathematical mistake
    * Incompleteness is not a mathematical error.
- EFFICIENT VERIFICATION APPROACH:
- 1. QUICK COMPARISON (Use this to focus your attention):
    * Immediately identify where the solutions differ in approach or results
    * Use these differences as "error hotspots" to prioritize your verification
    * When solutions agree, you can generally assume that part is correct
    * When solutions disagree, investigate those specific points deeply
- 2. TARGETED VERIFICATION (Only where needed):
    * Most important: Do not consider any incomplete steps as errors
    * Focus your mathematical verification on the "hotspots" identified above
    * Check mathematical validity only at points of difference or uncertainty
    * Avoid line-by-line checking of steps where solutions agree
    * For each potential error spot, verify if the mathematical reasoning is valid
    * If an intermediate step is later corrected, do not penalize the solution for having the incorrect intermediate step
- After your targeted verification, propose a score tuple (score_1, score_2):
    * Score (1,1) if both partial solutions are valid
    * Score (1,0) if only the first solution is valid
    * Score (0,1) if only the second solution is valid
    * Score (0,0) if both solutions are invalid
- In case you score a solution as 0, you must give an explanation for each check below:
    * If you score a solution as 0, you MUST identify the specific mathematical error.
    * You must also double check the problem statement. Reconsider your score and determine if you have misinterpreted the problem statement.
    * You must also check whether you have penalized a solution for being incomplete or having missing steps.
- Before outputting your final score, you must answer these questions:
    * STOP! Did you give a score of 0 to a solution that was incomplete?
    * STOP! Did you penalize a solution for being incomplete or having missing steps?
    * STOP! Did you make a comparison of which solution is better?
    * STOP! Did you consider steps incorrect just because they differ between solutions?
    * STOP! Did you prematurely evaluate based on final answers?
    * STOP! Did you consider a step incorrect just because it lacks sufficient detail or justification?

Now give your final score: Final score:

---

The prompt for synthesizing a final response from the pruned candidate response set using the same LM $\mathcal{M}$ is given below:

---

**Final synthesis prompt for guided self-verification**

Solve the following math problem with mathematical precision and clarity.
Problem: {problem}
Below are potential solution approaches with sections marked as uncertain (between `*START_UNCERTAIN_REGION*` and `*END_UNCERTAIN_REGION*`). These sections may contain conceptual or computational errors.
There are also sections marked as `*START_POSSIBLE_ERROR*` and `*END_POSSIBLE_ERROR*`. A verification step indicated that these steps are highly likely to contain errors.
Potential Approaches: {masked_candidate_responses}
  - Your task:
    1. Analyze all potential approaches critically, identifying their mathematical strengths and weaknesses If the approaches contain different answers, think carefully about why they are different, and use this to identify potential errors.
    2. Using the sections with special markers, identify potential errors.
    3. Develop a rigorous, step-by-step solution based on sound mathematical principles
    4. For uncertain regions:
        * Verify each step using algebraic or numerical validation
        * If correct, incorporate these steps with appropriate justification
        * If incorrect, provide clear corrections with mathematical reasoning for your changes
    5. Follow a comparative approach, using the differences between approaches to identify potential errors.
    6. Do not blindly follow the approaches, but rather use them to identify potential errors.
  - Guidelines for your solution:
    * Begin with a strategic overview of your chosen approach
    * Present each mathematical step with clear notation and justification
    * Pay special attention to areas that were previously marked uncertain
Conclude your solution with: Therefore, the final answer is: `$\boxed{{answer}}$`. Solution:

---

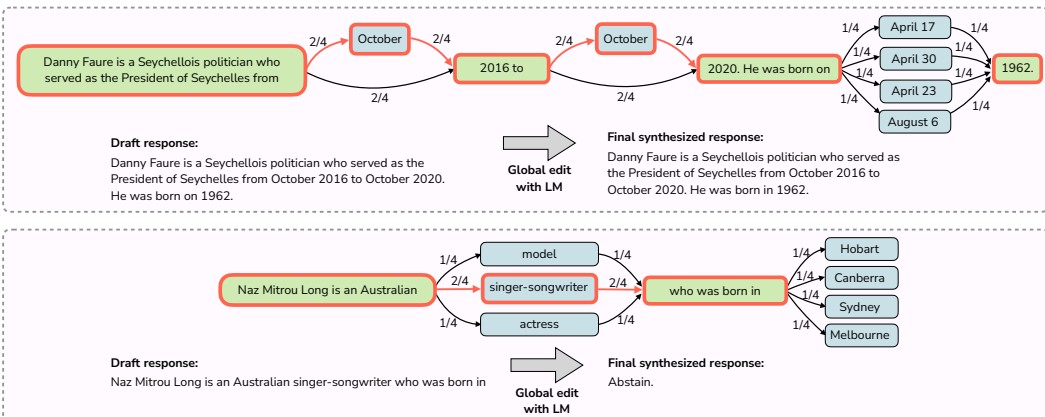

Figure 6: Consensus decoding synthesizes responses by traversing a CONGR and selecting nodes that are present in at least a $\tau$ fraction of responses, where $\tau \in [0, 1]$ is a hyperparameter ($\tau = 0.5$ in this example). The selected nodes' text labels are concatenated and processed by a secondary LM to produce a coherent response (top). When this is not possible, the result is an abstention (bottom).

# F  MANUAL ANNOTATION OF THE FINAL STEP IN CONSENSUS DECODING

Consensus decoding uses a secondary LM to remove potential disfluencies from a draft synthesized response. We use manual inspection to quantify the impact of this step. For each of 100 biography examples, we generate 5 responses from Qwen2.5-72B-Instruct. We then perform CONGRS construction followed by consensus decoding. We manually inspect all differences between the draft synthesized response and the final synthesized response produced by the secondary LM.

We find that five claims (one each across five responses) are erroneously dropped, two claims (one each across two responses) are modified, and two claims (one each across two responses) are added. A few examples lost short phrases that are clear from context: one example lost a 3-word phrase and a 4-word phrase, and two examples lost instances of the word "also." One example lost a 9-word phrase and a 10-word phrase, though both were sentence fragments that did not correspond to claims.

# G  EXPERIMENT PROMPTS AND DETAILS

**Computing infrastructure.** We used a cluster compute node with 5 NVIDIA RTX A6000 GPUs and a Macbook Pro with an Apple M1 Pro with 16 GB of RAM for analysis. We also used the OpenAI and TogetherAI APIs.

**Runtime** Data regarding the average runtime per evaluation example is provided in Table 6.

Table 5: Hyperparameters for response generation for factuality and refusal experiments.

|  | Qwen2.5-72B Llama-3.3-70B OLMo-2-32B | Qwen2.5-7B Llama-3.1-8B OLMo-2-7B |
|---|---|---|
| Quantization | 8-bit | n/a |
| Temperature | 0.9 | 0.9 |
| samples_per_prompt | 5 | 5 |
| max_new_tokens | 500 | 500 |
| Random seed | 42 | 42 |

## G.1  FACTUALITY AND REFUSAL-BASED EXPERIMENTS

**Generation hyperparameters.** We first generate five samples per test example that our method will synthesize for the factuality and refusal-based tasks. For the factuality tasks, we use the same prompts as Jiang et al. (2024) while for the refusal-based tasks, we use the same prompts as Ravichander et al. (2025). Table 5 gives our hyperparameters.

Table 6: Mean combined runtime per example for CONGR construction (§2) and decoding experiments (§4).

| Dataset | Time (s) |
|---|---|
| Biographies | 33 |
| PopQA | 38 |
| Scientific Attributions | 31 |
| False Presuppositions | 3 |
| Historical Events | 28 |
| MATH | 150 |
| AIME | 360 |

**Methods.** We compare consensus decoding with CONGRS against the following methods and baselines:

- *Consensus decoding with* CONGRS: We apply consensus decoding with selection thresholds of $\tau = 0.3, 0.5$. We also perform additional analyses for biography-based tasks with different selection thresholds in Appendix I.

- *Minimum Bayes Risk Decoding* Bertsch et al. (2023): performs pairwise comparisons among responses to choose the response with lowest expected risk. We use BERTScore (Zhang* et al., 2020) as the risk function.

- *LM consensus*: We prompt an LM to synthesize a consensus response, with an option to abstain if there is too much variation. Exact prompts are below This method is similar to Universal Self-Consistency (Chen et al., 2024c), except that we synthesize a new response instead of selecting one response.
  For the LM Consensus baseline, we use gpt-4o-mini for generating the consensus generation with an added option to abstain. We use the following prompt:

> **LM consensus baseline prompt**
>
> You are given 5 texts. Your task is to form/generate a consensus/agreement text using the given texts. Consensus or agreement would mean producing a new text that uses the given 5 texts to find a coherent text that includes words and information that is consistent across all the given texts. Text 1: text[0] Text 2: text[1] Text 3: text[2] Text 4: text[3] Text 5: text[4]
> Here are some important guidelines:
> - If the texts differ at a certain point/word, the consensus text should select the most frequent word from among the given texts at the point of difference.
> - If the texts differ at a certain point/word and there is no most frequent word, the consensus text should select the word that is most similar to the other words in the text.
> - Abstain if the texts are too different and no consensus can be reached.
> Strictly follow the guidelines above, especially regarding abstaining if the texts are too different.
> Return your generation in the following format. Do not include any other text:
> consensus text: [your consensus text here]

- *Greedy decoding*: We generate a single greedy response using a temperature of zero and the following hyperparameters: *do_sample*: False, *max_new_tokens*: 500.

- *Shortest response*: We prompt an LM to select the shortest response from the candidate set of multiple responses based on number of words in the response. It has been observed that correct responses are often shorter than incorrect responses for some tasks (Dimakis, 2025; Marjanović et al., 2025). We hypothesize that shorter responses may contain fewer hallucinations. We use gpt-4o-mini with the following prompt:

> **Shortest response baseline prompt**
>
> You are given 5 texts. Text 1: text[0] Text 2: text[1] Text 3: text[2] Text 4: text[3] Text 5: text[4]
> Your task is to output the shortest text amongst the given texts in terms of total words in a text. Compare the texts and select the one that is the shortest. If there are multiple texts with the same length, select the first one.
> Strictly follow the guidelines above.
> Return your generation in the following format. Do not include any other text:
> shortest text: [your shortest text here]

- *Reasoning model response*: We use QwQ-32B (Yang et al., 2024), a state-of-the-art reasoning model with thinking and reflection capabilities. We generate only one response, since we are using QwQ-32B to compare our method against *serial* inference-time scaling as opposed to parallel. Due to this, we use a larger context window of 4096 tokens. Note that this method does not utilize the original set of model responses. We use the following standard hyperparameters for QwQ 32B: *quantization*: 4-bit, *temperature*: 0.6, *top_k*: 40, *top_p*: 0.95, *do_sample*: True, *random seed*: 42, *samples_per_prompt*: 1, *max_new_tokens*: 4096.

**Evaluation.** We use FActScore to decompose the factuality task baseline responses into atomic units for verification against the Wikipedia database. In addition to FActScore, we also compute the number of supported, unsupported, and total facts for the response.

For the refusal-based tasks, we use the HALoGEN code for evaluation of response ratio and hallucination scores of the baseline responses. We use gpt-4.1-mini as an additional LM judge for assessing abstention in a given response with the following prompt:

> **Abstention prompt**
>
> You are given a prompt and its response . Your task is to judge whether the response to the prompt is an abstention from answering or not? Just answer with 'yes' or 'no'. 'yes' if it is an abstention, 'no' if it is not an abstention and it seems like an answer.
> prompt: prompt, response: response
> Return your generation in the following format. Do not include any other text:
> abstention: [your judgment here]

## G.2 REASONING EXPERIMENTS

For both MATH and AIME, we use the following configuration for both models: *quantization*: 8-bit, *temperature*: 0.9, *random seed*: 42, *samples_per_prompt*: 5

For MATH, we use *max_new_tokens* = 1024, whereas for the more difficult AIME dataset, we use *max_new_tokens* = 8192.

For both MATH and AIME, we use the following prompt to generate samples:

> **Prompt for generating samples for MATH and AIME**
>
> Solve the following math problem efficiently and clearly:
>
> - For simple problems (2 steps or fewer): Provide a concise solution with minimal explanation.
>
> - For complex problems (3 steps or more): Use this step-by-step format:
>
>   ## Step 1: [Concise description] [Brief explanation and calculations]
>   ## Step 2: [Concise description] [Brief explanation and calculations]
>   ...
>
> Regardless of the approach, always conclude with:
> Therefore, the final answer is: `$\boxed{{answer}}$`. I hope it is correct.
> Where {answer} is just the final number or expression that solves the problem.
> Problem: {problem}

We evaluate the following methods:

- *Guided Self-Verification with* CONGRS: We apply guided self-verification with pruning threshold of $\kappa = 0.7$.
- *Self-consistency* (Wang et al., 2023b): We take a majority vote of the final answer over all $m$ responses.
- *Self-verification* (Zhao et al., 2025; Tyen et al., 2024; Weng et al., 2023): We ask the same LM to score all $m$ responses based on correctness and select the best scoring response. For fairness, the verification prompt is kept the same as our guided self-verification method.
- *Pass@m*: We measure whether at least one of the $m$ responses contains a correct answer. This represents an upper bound for methods that aggregate over responses.

# H    Full Results

Table 8 and Table 9 give our main results for biography generation and PopQA. We also benchmark our factuality and refusal-based task baselines on three additional models including Qwen2.5-7B, Llama-3.1-8B, and OLMo-2-7B. We report results in Table 10, Table 11, and Table 13. We also explore the third refusal-based HALoGEN task, Historical Events. This task evaluates the extent of LM hallucination for historical events. We report results in Table 12.

Table 7 shows that when consensus decoding does abstain, it does so for entities where the original responses have low FActScore on average.

Table 7: Mean FActScore of original responses for entities that consensus decoding abstains on, compared to FActScore on all entities. Consensus decoding abstains on responses that are highly likely to contain hallucinations.

| $\tau$ | Model | FActScore Abstained Entities | FActScore All entities |
|---|---|---|---|
| 0.3 | Qwen2.5-72B | 0.31 | 0.68 |
| | Qwen2.5-7B | 0.34 | 0.68 |
| 0.5 | Qwen2.5-72B | 0.20 | 0.64 |
| | Qwen2.5-7B | 0.28 | 0.64 |

Table 8: Results for biography generation: FActScore, numbers of supported (#T) and unsupported (#F) facts, and response ratio (R, how often the model doesn't abstain). We report mean values over five replications. Standard deviation across runs is in small font.

| | Qwen2.5-72B | | | | Llama-3.3-70B | | | | OLMo-2-32B | | | |
|---|---|---|---|---|---|---|---|---|---|---|---|---|
| Method | FActScore | #T | #F | R | FActScore | #T | #F | R | FActScore | #T | #F | R |
| Greedy | 0.677 0.002 | 18.771 0.192 | 8.553 0.149 | 0.974 0.005 | 0.645 0.073 | 16.343 2.965 | 6.271 0.532 | 0.620 0.115 | 0.749 0.005 | 24.338 0.303 | 7.592 0.226 | 0.998 0.005 |
| Mean of $m$ | 0.681 0.005 | 19.209 0.146 | 8.455 0.156 | 0.966 0.007 | 0.641 0.007 | 16.420 0.565 | 6.487 0.129 | 0.589 0.016 | 0.740 0.014 | 25.958 0.274 | 7.764 0.384 | 0.976 0.006 |
| Shortest | 0.689 0.009 | 19.018 0.338 | 8.121 0.258 | 0.952 0.004 | 0.692 0.023 | 16.989 0.551 | 4.931 0.427 | 0.482 0.027 | 0.765 0.011 | 25.864 0.710 | 7.372 0.266 | 0.950 0.014 |
| LM consensus | 0.693 0.010 | 20.925 0.330 | 8.979 0.448 | 0.974 0.005 | 0.687 0.014 | 19.966 0.299 | 6.587 0.318 | 0.618 0.019 | 0.757 0.005 | 25.380 0.405 | 7.998 0.224 | 0.994 0.009 |
| MBR | 0.696 0.006 | 19.469 0.097 | 7.984 0.333 | 0.962 0.007 | 0.676 0.014 | 17.505 0.359 | 5.779 0.592 | 0.566 0.017 | 0.756 0.006 | 25.650 0.652 | 8.122 0.405 | 0.988 0.013 |
| QwQ-32B | 0.562 0.003 | 19.830 0.283 | 15.073 0.196 | 0.974 0.006 | 0.562 0.003 | 19.830 0.283 | 15.073 0.196 | 0.974 0.006 | 0.561 0.010 | 19.862 0.353 | 15.120 0.902 | 0.978 0.005 |
| ASC ($\Theta = 2$) | 0.688 0.008 | 18.766 0.332 | 6.128 0.174 | 0.968 0.008 | 0.715 0.031 | 13.179 4.619 | 3.367 0.282 | 0.578 0.017 | 0.761 0.012 | 18.734 1.226 | 4.208 0.197 | 0.954 0.021 |
| ASC ($\Theta = 3$) | 0.733 0.013 | 12.880 0.602 | 2.988 0.366 | 0.882 0.015 | 0.796 0.018 | 9.483 3.662 | 1.933 0.241 | 0.486 0.014 | 0.827 0.013 | 8.972 0.872 | 1.548 0.250 | 0.870 0.010 |
| CONGRS ($\tau$=0.3) | 0.715 0.014 | 20.784 0.379 | 6.007 0.509 | 0.924 0.008 | 0.827 0.005 | 22.226 1.120 | 2.722 0.264 | 0.462 0.023 | 0.815 0.027 | 26.768 1.490 | 4.578 0.270 | 0.924 0.011 |
| CONGRS ($\tau$=0.5) | 0.795 0.021 | 17.462 0.358 | 2.970 0.270 | 0.846 0.021 | 0.845 0.004 | 18.395 1.157 | 1.814 0.148 | 0.430 0.014 | 0.862 0.020 | 18.714 1.837 | 2.790 0.353 | 0.828 0.015 |

Table 9: Results for **PopQA**. We report FActScore (Min et al., 2023), number of supported (#T) and unsupported (#F) facts, and response ratio (R, how often the model responds and doesn't abstain). Like in Table 8, consensus decoding with $\tau = 0.3$ consistently improves FActScore by decreasing the number of unsupported facts. We report mean values over five replications. Standard deviation across runs is in small font.

| | Qwen2.5-72B | | | | Llama-3.3-70B | | | | OLMo-2-32B | | | |
|---|---|---|---|---|---|---|---|---|---|---|---|---|
| Method | FActScore | #T | #F | R | FActScore | #T | #F | R | FActScore | #T | #F | R |
| Greedy | 0.684 0.006 | 18.652 0.402 | 8.184 0.108 | 0.990 0.010 | 0.620 0.007 | 15.804 0.183 | 8.296 0.059 | 0.840 0.007 | 0.738 0.005 | 23.096 0.155 | 7.502 0.066 | 0.984 0.009 |
| Mean of $m$ | 0.694 0.006 | 18.720 0.705 | 8.096 0.104 | 0.956 0.011 | 0.626 0.015 | 15.548 0.538 | 8.720 0.209 | 0.814 0.015 | 0.746 0.006 | 24.354 0.226 | 7.838 0.073 | 0.982 0.011 |
| Shortest | 0.714 0.010 | 19.184 0.378 | 7.344 0.273 | 0.958 0.005 | 0.644 0.006 | 14.724 0.419 | 7.322 0.301 | 0.716 0.009 | 0.752 0.008 | 23.930 0.116 | 7.366 0.343 | 0.974 0.006 |
| LM consensus | 0.722 0.005 | 21.708 0.062 | 8.270 0.313 | 0.980 0.010 | 0.684 0.011 | 17.238 0.712 | 8.418 0.072 | 0.878 0.008 | 0.762 0.013 | 26.146 0.717 | 7.186 0.413 | 0.990 0.010 |
| MBR | 0.712 0.013 | 19.616 0.268 | 7.368 0.259 | 0.968 0.008 | 0.656 0.013 | 13.910 1.110 | 7.908 0.043 | 0.836 0.009 | 0.762 0.008 | 25.388 1.170 | 7.282 0.267 | 0.958 0.034 |
| QwQ-32B | 0.576 0.008 | 22.068 0.718 | 15.686 0.289 | 0.972 0.005 | 0.576 0.031 | 22.068 0.718 | 15.686 0.289 | 0.972 0.005 | 0.576 0.006 | 22.068 0.718 | 15.686 0.289 | 0.972 0.005 |
| ASC ($\Theta = 2$) | 0.724 0.006 | 17.384 0.158 | 5.698 0.138 | 0.976 0.006 | 0.712 0.016 | 15.390 0.903 | 5.502 0.283 | 0.728 0.005 | 0.728 0.016 | 17.336 1.063 | 5.256 0.199 | 0.918 0.008 |
| ASC ($\Theta = 3$) | 0.748 0.008 | 10.104 0.097 | 3.744 0.199 | 0.922 0.008 | 0.758 0.023 | 8.064 0.802 | 2.340 0.155 | 0.632 0.008 | 0.836 0.033 | 7.194 2.873 | 2.354 0.134 | 0.870 0.010 |
| CONGRS ($\tau$=0.3) | 0.742 0.009 | 19.298 0.183 | 5.882 0.141 | 0.968 0.005 | 0.760 0.012 | 16.962 0.377 | 4.784 0.181 | 0.682 0.005 | 0.792 0.019 | 22.924 0.774 | 5.622 0.343 | 0.886 0.009 |
| CONGRS ($\tau$=0.5) | 0.822 0.024 | 13.934 0.507 | 3.776 0.078 | 0.872 0.011 | 0.786 0.018 | 12.446 0.660 | 2.644 0.090 | 0.580 0.007 | 0.846 0.027 | 16.036 0.231 | 2.662 0.215 | 0.774 0.018 |

Table 10: Results for **Biography generation**. We report FActScore (Min et al., 2023), numbers of supported (#T) and unsupported (#F) facts, and response ratio (R, how often the model responds and doesn't abstain) for smaller models. Consensus decoding with $\tau = 0.3$ consistently improves FActScore by decreasing the number of unsupported facts. Using a threshold of $\tau = 0.5$ further improves FActScore by filtering more facts, at the expense of the total number of facts.

| Method | Qwen2.5-7B | | | | Llama-3.1-8B | | | | OLMo-2-7B | | | |
|---|---|---|---|---|---|---|---|---|---|---|---|---|
| | FActScore | #T | #F | R | FActScore | #T | #F | R | FActScore | #T | #F | R |
| Greedy | 0.68 | 18.40 | 8.27 | 0.98 | 0.85 | 15.24 | 2.28 | 0.72 | 0.60 | 22.71 | 14.03 | 1.00 |
| Mean of $m$ | 0.56 | 16.02 | 11.69 | 0.99 | 0.78 | 19.96 | 4.77 | 0.76 | 0.59 | 24.23 | 14.56 | 0.99 |
| Shortest | 0.57 | 15.81 | 11.13 | 0.99 | 0.81 | 20.21 | 4.10 | 0.71 | 0.58 | 22.35 | 13.80 | 0.98 |
| LM Consensus | 0.58 | 18.43 | 13.03 | 1.00 | 0.67 | 19.64 | 7.00 | 0.64 | 0.64 | 24.69 | 13.33 | 1.00 |
| MBR | 0.56 | 15.76 | 11.35 | 1.00 | 0.81 | 20.60 | 4.24 | 0.75 | 0.60 | 22.08 | 13.42 | 1.00 |
| QwQ-32B | 0.55 | 19.26 | 14.05 | 0.98 | 0.55 | 19.26 | 14.05 | 0.98 | 0.55 | 19.26 | 14.05 | 0.98 |
| CONGRS ($\tau$=0.3) | 0.64 | 16.75 | 6.21 | 0.87 | 0.80 | 21.26 | 3.86 | 0.74 | 0.78 | 26.29 | 4.86 | 0.70 |
| CONGRS ($\tau$=0.5) | 0.76 | 11.71 | 2.68 | 0.75 | 0.85 | 20.32 | 1.78 | 0.41 | 0.85 | 19.46 | 2.52 | 0.61 |

Table 11: Results for **PopQA**. We report FActScore (Min et al., 2023), number of supported (#T) and unsupported (#F) facts, and response ratio (R, how often the model responds and doesn't abstain) for smaller models. Like in Table 10, consensus decoding with $\tau = 0.3$ consistently improves FActScore by decreasing the number of unsupported facts.

| Method | Qwen2.5-7B | | | | Llama-3.1-8B | | | | OLMo-2-7B | | | |
|---|---|---|---|---|---|---|---|---|---|---|---|---|
| | FActScore | #T | #F | R | FActScore | #T | #F | R | FActScore | #T | #F | R |
| Greedy | 0.55 | 15.11 | 11.74 | 0.99 | 0.66 | 16.06 | 7.92 | 0.86 | 0.60 | 19.76 | 12.59 | 0.98 |
| Mean of $m$ | 0.52 | 14.33 | 12.69 | 0.99 | 0.65 | 16.11 | 8.27 | 0.82 | 0.59 | 20.36 | 13.82 | 0.98 |
| Shortest | 0.56 | 14.86 | 11.87 | 0.99 | 0.65 | 15.93 | 7.60 | 0.73 | 0.58 | 19.67 | 13.93 | 0.97 |
| LM Consensus | 0.61 | 17.67 | 10.97 | 0.96 | 0.72 | 20.25 | 7.77 | 0.87 | 0.70 | 23.47 | 9.84 | 0.99 |
| MBR | 0.51 | 13.91 | 12.47 | 0.96 | 0.69 | 17.06 | 7.19 | 0.83 | 0.62 | 20.33 | 12.30 | 0.98 |
| QwQ-32B | 0.57 | 22.37 | 15.81 | 0.97 | 0.57 | 22.37 | 15.81 | 0.97 | 0.57 | 22.37 | 15.81 | 0.97 |
| CONGRS ($\tau$=0.3) | 0.66 | 11.33 | 4.78 | 0.83 | 0.74 | 13.11 | 4.04 | 0.71 | 0.75 | 18.18 | 4.86 | 0.83 |
| CONGRS ($\tau$=0.5) | 0.76 | 7.74 | 2.12 | 0.73 | 0.82 | 10.83 | 2.47 | 0.60 | 0.86 | 10.09 | 1.85 | 0.54 |

Table 12: Performance on the **Historical Events** task from HALoGEN (Ravichander et al., 2025). We report Response Ratio (R↓) and Hallucination Score (H↓), both lower is better. Consensus decoding with CONGRS consistently achieves low Response Ratio and Hallucination Score for the OLMo model family.

| Method | Qwen2.5-72B | | Qwen2.5-7B | | Llama-3.3-70B | | Llama-3.1-8B | | OLMo-2-32B | | OLMo-2-7B | |
|---|---|---|---|---|---|---|---|---|---|---|---|---|
| | R↓ | H↓ | R↓ | H↓ | R↓ | H↓ | R↓ | H↓ | R↓ | H↓ | R↓ | H↓ |
| Greedy | 0.008 | 0.008 | 0.064 | 0.064 | 0 | 0 | 0 | 0 | 0.280 | 0.280 | 0.004 | 0.004 |
| Mean of $m$ | 0.008 | 0.008 | 0.056 | 0.056 | 0.0008 | 0.0008 | 0.0032 | 0.0032 | 0.252 | 0.252 | 0.0064 | 0.0064 |
| Shortest | 0.008 | 0.008 | 0.024 | 0.024 | 0 | 0 | 0 | 0 | 0.148 | 0.148 | 0.004 | 0.004 |
| LM Consensus | 0.012 | 0.012 | 0.056 | 0.056 | 0.004 | 0.004 | 0 | 0 | 0.312 | 0.312 | 0.004 | 0.004 |
| MBR | 0.008 | 0.008 | 0.040 | 0.040 | 0.004 | 0.004 | 0 | 0 | 0.200 | 0.200 | 0.004 | 0.004 |
| QwQ-32B | 0.232 | 0.232 | 0.232 | 0.232 | 0.232 | 0.232 | 0.232 | 0.232 | 0.232 | 0.232 | 0.232 | 0.232 |
| CONGRS ($\tau$=0.3) | 0.008 | 0.008 | 0.028 | 0.028 | 0.004 | 0.004 | 0.016 | 0.016 | 0.096 | 0.096 | 0 | 0 |
| CONGRS ($\tau$=0.5) | 0.008 | 0.008 | 0.012 | 0.012 | 0 | 0 | 0.008 | 0.008 | 0.060 | 0.060 | 0 | 0 |

Table 13: Performance on the **False Presuppositions** and **Scientific References** tasks from HALoGEN when synthesizing responses from small model sizes. We report Response Ratio (R↓) and Hallucination Score (H↓), both lower is better. Consensus decoding with CONGRS consistently achieves low Response Ratio and Hallucination Score.

| | False Presuppositions | | | | | | Scientific References | | | | | |
|---|---|---|---|---|---|---|---|---|---|---|---|---|
| | Qwen2.5-7B | | Llama-3.1-8B | | OLMo-2-7B | | Qwen2.5-7B | | Llama-3.1-8B | | OLMo-2-7B | |
| Method | R↓ | H↓ | R↓ | H↓ | R↓ | H↓ | R↓ | H↓ | R↓ | H↓ | R↓ | H↓ |
| Greedy | 0.40 | 0.28 | 0.60 | 0.45 | 0.51 | 0.41 | 0.56 | 0.49 | 0.42 | 0.33 | 0.89 | 0.81 |
| Mean of $m$ | 0.41 | 0.30 | 0.61 | 0.47 | 0.51 | 0.42 | 0.67 | 0.62 | 0.49 | 0.43 | 0.88 | 0.82 |
| Shortest | 0.36 | 0.25 | 0.50 | 0.34 | 0.54 | 0.46 | 0.49 | 0.45 | 0.29 | 0.25 | 0.77 | 0.71 |
| LM Consensus | 0.48 | 0.36 | 0.69 | 0.55 | 0.76 | 0.63 | 0.05 | 0.04 | 0.15 | 0.13 | 0.05 | 0.04 |
| MBR | 0.39 | 0.28 | 0.58 | 0.43 | 0.48 | 0.39 | 0.65 | 0.61 | 0.46 | 0.39 | 0.91 | 0.84 |
| QwQ-32B | 0.04 | 0.01 | 0.04 | 0.01 | 0.04 | 0.01 | 0.83 | 0.74 | 0.83 | 0.74 | 0.83 | 0.74 |
| CONGRS ($\tau$=0.3) | 0.10 | 0.06 | 0.09 | 0.05 | 0.13 | 0.10 | 0.07 | 0.06 | 0.30 | 0.26 | 0.15 | 0.13 |
| CONGRS ($\tau$=0.5) | 0.08 | 0.06 | 0.07 | 0.04 | 0.11 | 0.08 | 0.02 | 0.02 | 0.18 | 0.16 | 0.09 | 0.08 |

Table 14: Comparison of consensus decoding performance with beam search variants (all decoding-time algorithms). Results are using Qwen2.5-72B on biography generation. Consensus decoding with $\tau = 0.3$ outperforms all variants in terms of FActScore and number of supported/unsupported claims, with only a minor decrease in response ratio.

| Method | FActScore | #T | #F | R |
|---|---|---|---|---|
| Beam Search | 0.68 | 18.31 | 8.10 | 0.97 |
| Diverse Beam Search | 0.67 | 18.65 | 8.30 | 0.98 |
| Range Voting (Beam Search) | 0.69 | 19.39 | 8.14 | 0.97 |
| ConGrs ($\tau = 0.3$) | 0.70 | 20.81 | 6.97 | 0.93 |
| ConGrs ($\tau = 0.5$) | 0.77 | 17.48 | 3.48 | 0.85 |

# I  ABLATION EXPERIMENTS

## I.1  TEMPERATURE

For the biography generation setting with Qwen2.5-72B, we generate responses with different temperatures. CONGRS are effective at synthesizing information across responses even when the responses were generated with various temperatures.

Table 15: Ablation results for Qwen2.5-72B at different temperatures and CONGRS thresholds $\tau =$ 0.3, 0.5. We report FActScore, number of supported (#T) and unsupported facts (#F), and response ratio (R) averaged across the 100 entities.

| Temperature | Threshold $\tau$ | FActScore | #T | #F | R |
|---|---|---|---|---|---|
| 1.2 | CONGRS (0.3) | 0.74 | 20.58 | 5.25 | 0.89 |
|  | CONGRS (0.5) | 0.84 | 16.20 | 2.65 | 0.80 |
| 1.4 | CONGRS (0.3) | 0.75 | 21.54 | 5.25 | 0.88 |
|  | CONGRS (0.5) | 0.84 | 16.98 | 2.35 | 0.81 |
| 1.6 | CONGRS (0.3) | 0.76 | 21.45 | 5.43 | 0.86 |
|  | CONGRS (0.5) | 0.82 | 16.35 | 2.57 | 0.79 |
| 1.8 | CONGRS (0.3) | 0.77 | 21.44 | 4.65 | 0.82 |
|  | CONGRS (0.5) | 0.87 | 16.03 | 1.83 | 0.77 |
| 2.0 | CONGRS (0.3) | 0.78 | 21.78 | 4.35 | 0.83 |
|  | CONGRS (0.5) | 0.88 | 16.28 | 1.65 | 0.75 |

## I.2  NUMBER OF RESPONSES

We generate $m = 10$ responses per entity, for a set of 25 randomly sampled entities for the biography generation setting with Qwen2.5-72B; with temperatures 0.7 and 0.9. We report results for our consensus decoding method with different threshold $\tau$ values. CONGRS are effective at synthesizing information across 10 responses.

Table 16: Ablation results for Qwen2.5-72B ($m = 10$) at different temperatures and CONGRS thresholds $\tau$. We report FActScore, number of supported (#T) and unsupported facts (#F), and response ratio (R) averaged across the 25 sampled entities.

| Temperature | Threshold $\tau$ | FActScore | #T | #F | R |
|---|---|---|---|---|---|
|  | CONGRS (0.1) | 0.47 | 20.80 | 30.36 | 1.00 |
|  | CONGRS (0.2) | 0.49 | 16.72 | 14.52 | 1.00 |
| 0.9 | CONGRS (0.3) | 0.67 | 13.86 | 4.81 | 0.84 |
|  | CONGRS (0.4) | 0.75 | 11.90 | 2.90 | 0.84 |
|  | CONGRS (0.5) | 0.79 | 11.85 | 2.20 | 0.80 |
|  | CONGRS (0.7) | 0.81 | 9.58 | 1.74 | 0.76 |
|  | CONGRS (0.1) | 0.59 | 25.44 | 22.44 | 1.00 |
|  | CONGRS (0.2) | 0.59 | 21.56 | 12.56 | 1.00 |
| 0.7 | CONGRS (0.3) | 0.67 | 20.41 | 7.68 | 0.88 |
|  | CONGRS (0.4) | 0.70 | 18.45 | 5.23 | 0.88 |
|  | CONGRS (0.5) | 0.74 | 16.45 | 4.05 | 0.88 |
|  | CONGRS (0.7) | 0.78 | 13.70 | 2.30 | 0.80 |

### I.3 EXTERNAL LLM JUDGE

We experiment with using a small, open source model as the LM judge for CONGR construction and consensus decoding for biography generation. We use Qwen2.5-72B as the response generator and Qwen3-8B as the external LM judge. All other relevant parameters are fixed. Results are in Table 17, and show that our method is robust to using a less powerful LM judge and are within the original error margins.

Table 17: Ablation results for Qwen2.5-72B comparing different LM judges. We report FActScore, number of supported (#T) and unsupported facts (#F), and response ratio (R) averaged across the 100 entities.

| LM Judge | FActScore | #T | #F | R |
|---|---|---|---|---|
| CONGRS (0.3, Qwen3-8B Judge) | 0.70 | 20.73 | 6.95 | 0.93 |
| CONGRS (0.5, Qwen3-8B Judge) | 0.76 | 17.44 | 3.50 | 0.93 |
| CONGRS (0.3) | 0.70 | 20.81 | 6.97 | 0.93 |
| CONGRS (0.5) | 0.77 | 17.48 | 3.48 | 0.85 |

### I.4 SELECTION THRESHOLD

**Selection threshold $\tau$ trades off informativeness and factuality.** The frequency of each entity in pre-training data varies across our sets of entities. As a result, the factuality of the models' original responses varies as well. In the case of rare entities, variation between model responses can be indicative of hallucinations. However, for very common entities, variation may not be a result of hallucinations. In general, the threshold $\tau$ that we choose when performing consensus decoding controls whether the final result is more of an intersection between the original set of responses or whether it is a union of the responses.

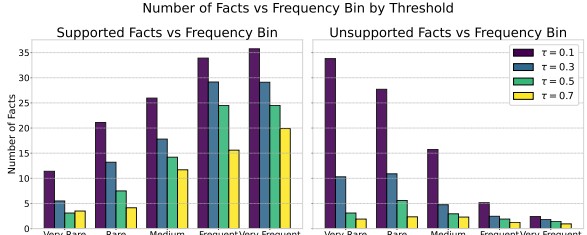

Figure 7: Different selection thresholds for Consensus Decoding are better for different kinds of entities. Generated biographies for very frequent entities are often very factual and benefit from a permissive, low value of $\tau$. On the other hand, generated biographies for very rare entities benefit from a high value of $\tau$, which only aggregates spans of text that occur in the vast majority of responses.

To study this trade-off between informativeness and factuality, we first estimate the frequency of each entity. Since we do not have access to each model's pretraining data, we use the monthly page views of each entity's Wikipedia page as a proxy measure of its frequency, as in Min et al. (2023). We then partition the entities into 5 equal-sized bins and perform consensus decoding with thresholds of $\tau = 0.1, 0.3, 0.5, 0.7$. We then analyze FActScores and the number of supported/unsupported claims for entities in each bin in Figure 7.

For very frequent entities, we see that all thresholds result in high FActScores. Moreover, even using the most permissive threshold $\tau = 0.1$ results in a negligible increase in the number of unsupported claims, while greatly increasing the number of supported claims.

