# OpenReview forum: "Sample, Align, Synthesize: Graph-Based Response Synthesis with ConGrs"
_ICLR.cc/2026/Conference — Submitted to ICLR 2026_

### Official Review · Reviewer_4yjc · 2025-10-31

**Soundness:** 3
**Presentation:** 3
**Contribution:** 2
**Rating:** 4
**Confidence:** 4

**Summary:**

This paper introduces a consensus-based decoding approach designed to enhance the performance of LLMs. The method first samples multiple responses from an LLM, constructs a DAG to represent their relationships, and then synthesizes them into a final, unified response. By leveraging this structured representation, the LLM can select and integrate complementary information from different outputs to produce a more coherent and accurate result. Experimental results demonstrate the effectiveness and robustness of the proposed method.

**Strengths:**

1. This paper presents a clear motivation and proposes a simple yet effective method for enhancing the performance of LLMs.

2. The process of constructing and utilizing the DAG is well presented, and the graph can be applied in various ways, such as aggregation and intervention.

3. Extensive experiments are conducted, and the results appear promising.

**Weaknesses:**

1. There is a considerable body of research on achieving consensus among multiple responses. However, key variants of self-consistency methods are missing from the comparison, which makes it difficult to fully assess the contribution of this work.

2. Although the constructed DAG is relatively simple, the draft response sampled from it essentially represents the shared content among multiple responses. It remains unclear how this approach fundamentally differs from existing methods in the literature, e.g., self-consistency.

3. As mentioned, the DAG-based approach is claimed to be cost-effective, as shown in Table 2. However, without comparisons against additional baselines, this claim is not well supported. Since only one baseline is used as a reference, it is difficult to determine whether the method is indeed cost-effective.

[1] https://aclanthology.org/2024.acl-long.634.pdf

**Questions:**

See Weakness

---

> ### Author Response · Authors · 2025-11-22
> **Author Response Part 1**
>
> Thanks for your review!
>
> ***Weakness 1**: There is a considerable body of research on achieving consensus among multiple responses. Key variants of self-consistency methods are missing from the comparison.*
>
> Thank you for this suggestion! Our method is different from self-consistency and its existing variants. Consider the long-form factuality setting.
> The original **self-consistency** [1] performs majority voting over short final answers (such as single phrases, multiple choice answers, and numbers), and as such is not applicable in the long-form factuality setting, where full responses are evaluated.
> **Universal self-consistency** [2]  uses an LLM judge to select the single “most consistent” response.
> **Fine-grained self-consistency** [3]  prompts an LLM judge to extract and combine parts of multiple responses in a black-box way. This method is very similar to the LLM consensus baseline in our paper, which our method consistently outperforms.
> **Atomic self-consistency** [4] (which we compare to in the paper) is, in our opinion, the most relevant self-consistency variant. It combines sentences from different responses. Figure 5 shows that our method is consistently better than this method at all parameter settings across two factuality tasks and three models.
> **Unigram Consistency Score** [5] is similar to MBR, and uses unigram overlap to select the response in a set that is most consistent with the other responses.
> Our **consensus decoding** method combines text spans of *variable lengths* from different responses–some spans will be as short as single words, while others can be multiple sentences long.
>
> We have provided new comparisons to FSC, USC, and UCS below on biography generation using Qwen 2.5 72B as the response generator. We find that our method is still on the Pareto frontier of factuality and informativeness. We have also incorporated more context about how our method differs from existing works in the Related Work section of our paper. We will also incorporate these results into the main paper itself.
>
> We also already compare to a claim-decomposition response-synthesis method that is not derived from self-consistency. **Uncertainty-aware decoding** uses an LLM to decompose all responses into a set of atomic claims. A bipartite graph of responses and claims is constructed and used to select which claims to combine with an external LLM. Our method achieves comparable performance while using 82% fewer tokens from external LLMs (Table 2).
>
> For good measure, we also already compare to direct decoding methods that are not derived from self-consistency in Table 14 in Appendix H:
> **Minimum-bayes risk decoding** selects the response that is most similar to every other response in the set. We use BERTScore to measure similarity.
> **Beam search** and variants: **Diverse beam search** [6], **Range voting** [7]
> **Uncertainty-aware decoding** uses an LLM to decompose all responses into a set of atomic claims. A bipartite graph of responses and claims is constructed and used to select which claims to combine with an external LLM. Our method achieves comparable performance while using 82% fewer tokens from external LLMs (Table 2).
>
> | Method | FActScore (&uarr;) | # supported claims (&uarr;) | # unsupported claims (&darr;) | response ratio |
> |-----|---:|--:|--:|--:|
> | FSC | 0.68 | 22.96 | 10.41 | 0.98 |
> | UCS | 0.68 | 18.46 | 8.40 | 0.98 |
> | USC | 0.70 | 19.85 | 8.15 | 0.98 |
> | ConGrs (tau = 0.3) | 0.70 | 20.81 | 6.97 | 0.93 |
> | ConGrs (tau = 0.5) | 0.77 | 17.48 | 3.48 | 0.85 |
>
> [1] Self-consistency improves chain of thought reasoning in language models. Wei et al., 2022.
>
> [2] Universal Self-Consistency for Large Language Models. Chen et al.,  ICML 2024 Workshop on In-Context Learning.
>
> [3]Integrate the Essence and Eliminate the Dross: Fine-Grained Self-Consistency for Free-Form Language Generation. Wang et al., ACL 2024.
>
> [4] Atomic Self-Consistency for Better Long Form Generations, Thirukovalluru et al., EMNLP 2024.
>
> [5]  Lightweight reranking for language model generations, Jain et al., ACL 2024.
>
> [6]: Vijayakumar et al., Diverse Beam Search: Decoding Diverse Solutions from Neural Sequence Models, AAAI 2018.
>
> [7]: Borgeaud and Emerson, Leveraging Sentence Similarity in Natural Language Generation: Improving Beam Search using Range Voting, Workshop on Neural Generation and Translation at ACL, 2020.

---

> > ### Author Response · Authors · 2025-11-22
> > **Author Response Part 2**
> >
> > ***Weakness 2**: The draft response sampled from the constructed DAG essentially represents the shared content among multiple responses. It remains unclear how this approach fundamentally differs from existing methods in the literature, e.g., self-consistency.*
> >
> > We provide additional information about how our method differs from existing variants, in addition to the information provided in the response to the previous weakness.
> >
> > A fundamental way our method differs from existing self-consistency variants is the fact that it can operate on a span level. In contrast, existing variants operate on full responses or fixed units such as claims or sentences, which limits flexibility.
> >
> > To illustrate this, consider the following set of examples (from the Biographies dataset we test on):
> >
> > 1. Vance Joy, born James Keogh on January 28, 1987, in Melbourne, Australia, is an acclaimed singer-songwriter…
> >
> > 2. Vance Joy, born James Keogh on January 21, 1987, in Melbourne, Australia, is an acclaimed singer-songwriter…
> >
> > 3. Vance Joy, born James Keogh on December 29, 1987, in Melbourne, Australia, is an acclaimed singer-songwriter…
> >
> > 4. Vance Joy, born James Keogh on December 27, 1987, in Melbourne, Australia, is an acclaimed singer-songwriter…
> >
> > Here, all responses share stable spans about the entity such as his birthplace and job title. However, they all disagree on a small span that includes the month and day, but not the year, of his birthday. In fact, all three full birthdates are incorrect, but all include the correct year. Existing methods relying on selection will necessarily include one of the full birthdates. Methods that rely on sentence/claim decomposition will also either include a full birthday or include none of it. Our method, on the other hand, can flexibly encode that all responses diverge on the day and month and not on the year. The resulting synthesized response will therefore include only the year, which happens to also be correct. This example illustrates one of the key advantages of our method.
> >
> > ***Weakness 3**: As mentioned, the DAG-based approach is claimed to be cost-effective, as shown in Table 2. However, without comparisons against additional baselines, this claim is not well supported.*
> >
> > We directly compare cost (in terms of tokens used from external LMs) with UAD in Table 2 because it is the method most similar to ours and one that uses external LM calls. Other methods like ASC combine responses using embedding models rather than external LMs, so it is not feasible to make a direct cost comparison. While this means that ASC uses fewer external LM tokens by default, our method offers a strictly higher factuality and informativeness in all scenarios that we tested (Figure 5). Compared to UAD, our method achieves similar results using 80% fewer tokens from external LMs (Table 2). Additionally, our method is more flexible than methods like ASC that do not use external LMs. Our approach can flexibly apply to a variety of tasks, including reasoning, since it doesn’t require claim or sentence-level decomposition.

---

> > > ### Comment · Area_Chair_qmu1 · 2025-11-22
> > >
> > > Hi Reviewer,
> > >
> > > The authors have submitted their responses to your reviews. Please take a look and let the authors know if you have any further questions or concerns. Thank you again for your contributions to ICLR!
> > >
> > > Best regards,
> > > AC

---

### Official Review · Reviewer_EiKb · 2025-10-31

**Soundness:** 3
**Presentation:** 3
**Contribution:** 3
**Rating:** 6
**Confidence:** 3

**Summary:**

The paper introduces Consensus Graphs (CONGRS), a novel graph-based data structure designed to generate a single, reliable response from multiple long-form outputs produced by a Large Language Model (LLM). The approach proceeds in three stages. First, multiple responses are sampled from the LLM. Second, these responses are aligned by constructing a CONGR, which identifies shared anchor spans (consensus nodes) and uses a secondary language model to assess semantic equivalence within the variable segments (disagreement nodes). Finally, a unified response is synthesized from this graph using one of two task-specific decoding strategies: Consensus Decoding (an aggregation method for factual tasks) or Guided Self-Verification (an intervention method for reasoning tasks).

**Strengths:**

1. The method is novel.
2. CONGRS-based synthesis provides significant gains in all areas such as: factuality, safety, and reasoning
3. The process is cost-efficient.

**Weaknesses:**

1. The CONGRS construction process fundamentally relies on the presence of anchor spans—identical lexical subsequences shared across multiple responses. The paper notes that this assumption holds well for aligned models but degrades in more open-ended tasks. In 6 out of 100 biography examples, the responses exhibited "so much variability that no consensus nodes are made."" For creative writing tasks, the consensus nodes contained "much less text," leading the authors to acknowledge that "CONGRS may offer more limited utility in that setting."
2. While the method is cost-effective, it is not cost-free and still relies on a secondary language model for two critical steps. An LM judge is required to "determine which parts of the responses between the anchor spans are semantically equivalent" , and another LM call is needed in the final synthesis step to "fix grammatical errors" and coherence. This introduces external dependencies and potential points of failure, for example if the judge LLM have bias and could be wrong (the entire process fail).
3. The paper claims that the final LM-based synthesis step, "does not introduce or remove hallucinations". However, this claim seems really weak when supported by a "qualitative analysis" and "manual inspection" of only "25 biography examples".

**Questions:**

1. The paper’s core alignment method depends on identifying anchor spans (consensus nodes) through lexical alignment. This approach performs well for aligned models but degrades on more open-ended or creative tasks. How robust is it to different kinds of alignments? For instance, if a model is aligned via RLHF techniques that promote lexical diversity, could this disrupt the lexical alignment process and lead to failures in forming consensus nodes, as observed in 6 of the 100 biography examples?
2. How sensitive is the final graph quality and response factuality to the strength of this LM judge? What would happen if you used a less-capable open-source model as the judge?
3. It seems to me that adaptive consensus threshold ($\tau$) requires manual tuning, since the paper shows that the optimal consensus threshold $\tau$ depends on the entity's rarity (rare entities need a higher, more conservative $\tau$). Can this be truely adaptive?
4. Your "Guided Self-Verification" method uses the CONGRS graph to localize errors for the LM to self-check. However, the paper cites that LMs "struggle to localize errors". So, how often does the LM fail to correct an error you've localized?

---

> ### Author Response · Authors · 2025-11-22
> **Author Response (Part 1)**
>
> Thanks for your review!
>
> ***Weakness 1**: The CONGRS construction process fundamentally relies on the presence of anchor spans—identical lexical subsequences shared across multiple responses. The paper notes that this assumption holds well for aligned models but degrades in more open-ended tasks.*
>
> We agree that ConGrs may not be as effective for tasks such as creative writing, where the frequency of anchor spans is diminished. However, ConGrs are designed for tasks where variation across samples carries epistemic signals. Creative or highly open-ended generations necessarily encourage stylistic and content variation. Divergences between responses for these tasks do not carry signals about, for instance, competing factual claims or alternative reasoning paths. We have addressed this more clearly in the paper (line 224, at the end of Section 2) to provide further guidance about what type of tasks ConGrs may be useful for, in addition to those we already experiment with.
>
> ***Weakness 2**: The LLM judge used in graph construction and consensus decoding introduces external dependencies and potential points of failure (e.g. bias, errors).*
>
> We agree that external LLM judges introduce a new source of error. This is a major reason why we use an LLM judge in very narrowly scoped use cases. The first is to determine whether two spans convey the same information during graph construction, and the second is to clean up disfluencies in a draft response during consensus decoding. In the response to the next weakness provided below, we provide additional analysis regarding the LLM judge’s influence on the consensus decoding generations and find that it is minimal.
>
> Unlike prior work, we do not use judges for more open-ended steps such as claim extraction, which are more complex and introduce greater risks of new bias and errors.
>
> ***Weakness 3**: The paper claims that the final LM-based synthesis step, "does not introduce or remove hallucinations". However, this claim seems really weak when supported by a "qualitative analysis" and "manual inspection" of only "25 biography examples".*
>
> We expanded our analysis to 100 biography examples from the biographies set using Qwen 2.5 72B. Out of these instances, we find 5 instances where the final global edit step removes a full non-hallucinated claim. We find 2 instances where the global edit step modifies a claim (changing an age from 38 to 39 and changing the name of a movie title). For context, there are roughly 30 claims per generation in this setting. We have updated the paper to reflect this (Appendix F and line 264).
>
> ***Question 1**: The lexical alignment approach performs well for aligned models but degrades on more open-ended or creative tasks. How robust is it to different kinds of alignments?*,
>
> This is a very interesting question. We first note that our results are over three model classes (OLMo, Llama, and Qwen), each with its own post-training procedures. In addition, we have run an experiment with a model using Creative Preference Optimization (CPO) [1], a recently published alignment method that explicitly aims to encourage diversity. We use the CrPO-llama-3.1-8b-instruct-div model that was released by the authors and test on our biography generation task (with no hyperparameter changes otherwise). In the table below, we report the average factuality of the original responses across three temperature settings. We also report factuality results after consensus decoding with ConGrs (with $\tau=0.3$. Even in this setting, ConGrs improve the factuality of responses. However, we note that the initial generations from the model trained with CPO are far less factual than responses from Llama 3.1 8B instruct, which we report in Table 8 in Appendix H of the paper. This suggests that there is a tradeoff between creativity and factuality when determining which alignment method to use.
>
> | Method | FActScore (&uarr;) | # supported claims (&uarr;) | # unsupported claims (&darr;) | response ratio |
> |-----|---:|--:|--:|--:|
> | CPO (temp=0.9) | 0.42  | 17.3  | 23.1 | 1.0 |
> | CPO (temp=0.5) | 0.50  | 20.69 | 20.59 | 1.0 |
> | CPO (temp=0.3) | 0.54 | 22.5 | 19.5 | 1.0 |
> | CPO (temp=0.9, CD) | 0.71 | 4.50 | 0.75 | 0.08 |
> | CPO (temp=0.5, CD) | 0.73 | 7.91 | 1.97 | 0.35 |
> | CPO (temp=0.3, CD)| 0.74 | 9.41 | 3.05 | 0.61 |
>
> [1] Creative preference optimization. Ismayilzada et al., EMNLP 2025.

---

> > ### Author Response · Authors · 2025-11-22
> > **Author Response Part 2**
> >
> > ***Question 2**: How sensitive is the final graph quality and response factuality to the strength of this LM judge? What would happen if you used a less-capable open-source model as the judge?*
> >
> > We have run new experiments testing the use of a different LM judge. In particular, we use Qwen 3 8B as a judge model (a relatively small, open model)  and repeat our experiments with Qwen 2.5 72B as the response set generator. As seen in the results below, the change in results is negligible. Moreover, we confirmed that they are within the original error margins. We have added these results to Appendix I of the paper.
> >
> > | Method | FActScore (&uarr;) | # supported claims (&uarr;) | # unsupported claims (&darr;) | response ratio |
> > |-----|---:|--:|--:|--:|
> > | ConGrs (tau = 0.3, Qwen Judge) | 0.70 | 20.73 | 6.95 | 0.93 |
> > | ConGrs (tau = 0.5, Qwen Judge) | 0.76 | 17.44 | 3.50 | 0.93 |
> > | ConGrs (tau = 0.3) | 0.70 | 20.81 | 6.97 | 0.93 |
> > | ConGrs (tau = 0.5) | 0.77 | 17.48 | 3.48 | 0.85 |
> >
> > ***Question 3**: The consensus threshold $\tau$ seems to require manual tuning for consensus decoding.*
> >
> > The hyperparameters for both of our decoding methods reflect user preferences about conservativeness rather than task-specific tuning. For instance, the consensus decoding threshold parameter $\tau$ controls the tradeoff between factual precision and recall. If the user has a low tolerance for factual errors, they can select a higher value of $\tau$ and vice versa. In either case, our method is on the Pareto frontier of factual precision and number of claims across all settings.
> >
> > ***Question 4**: Guided Self-Verification uses the CONGRS graph to localize errors for the LM to self-check, but the paper cites that LMs "struggle to localize errors". So, how often does the LM fail to correct an error you've localized?*
> >
> > We’d like to clarify that LMs struggle to localize errors without extra guidance. GSV is designed to use the structure of a ConGr to provide this guidance. In Table 4, we compare against a baseline that uses the same pairwise comparison that GSV relies on, with the difference being the fact that the model is given the full response without any kind of guidance as to where the possible errors could be. The lower performance in this setting demonstrates that the benefit provided by GSV is localization provided by the graph.

---

> > > ### Comment · Area_Chair_qmu1 · 2025-11-22
> > >
> > > Hi Reviewer,
> > >
> > > The authors have submitted their responses to your reviews. Please take a look and let the authors know if you have any further questions or concerns. Thank you again for your contributions to ICLR!
> > >
> > > Best regards,
> > > AC

---

### Official Review · Reviewer_VmdJ · 2025-11-01

**Soundness:** 3
**Presentation:** 3
**Contribution:** 2
**Rating:** 4
**Confidence:** 3

**Summary:**

This paper proposes consensus graphs to capture semantic variation across multiple LM-generated responses to the same prompt. By leveraging these graphs, the approach aims to synthesize more effective responses and enhance factual accuracy.

**Strengths:**

The method provides a reasonable way to identify uncertain segments in LM outputs, which can help mitigate some hallucinations and improve reliability.

**Weaknesses:**

1. The approach primarily addresses cases where the LM exhibits uncertainty. It is less effective when the LM is confidently incorrect, as consensus cannot correct such errors.
2. The method relies on multi-pass generation and semantic equivalence class identification, which introduces significant LM inference cost. Alternative approaches that utilize internal model states for improving truthfulness exist and may offer more efficiency. Including a comparison with such methods would strengthen the paper.

**Questions:**

1. Can this method be extended to achieve consensus across responses from multiple models, rather than a single LM?
2. How are hyperparameters selected? Do they require task-specific or model-specific tuning?

---

> ### Author Response · Authors · 2025-11-22
> **Author Response**
>
> Thanks for your review!
>
> ***Weakness 1**: The approach primarily addresses cases where the LM exhibits uncertainty. It is less effective when the LM is confidently incorrect, as consensus cannot correct such errors.*
>
> We agree that our method will not be as effective when the LM is confidently incorrect. There is always going to be some irreducible error, and any method that does not use external verification will face this issue. This kind of failure affects all methods that aggregate across multiple LM responses, including self-consistency and its variants, which our method outperforms. However, our results demonstrate that in general, agreement and disagreement across responses provide useful signals.
>
> ***Weakness 2**: The method relies on multi-pass generation and semantic equivalence class identification, which introduces significant LM inference cost. Alternative approaches that utilize internal model states for improving truthfulness exist and may offer more efficiency. Including a comparison with such methods would strengthen the paper.*
>
> Our method is designed to work with black-box LM access, while methods that utilize internal model states necessarily require white-box access. Our method also does not require any probe training which white-box methods, like [1], typically require. With that said, we do already compare our method against several methods that do not require external LMs, such as decoding with temperature 0 and MBR decoding (results in Figure 3 and Table 3). We also compare to several beam search variants in Table 14 of Appendix H. Our method is consistently on the pareto frontier of factuality and informativeness compared to these methods.
>
> [1]: Inference-Time Intervention: Eliciting Truthful Answers from a Language Model, Li et al., NeurIPS 2023.
>
> ***Question 1**: Can this method be extended to achieve consensus across responses from multiple models, rather than a single LM?*
>
> Our method as it stands is not optimized for aggregation across multiple models. It relies on the existence of shared anchor spans, which are present in responses sampled from the same model [1]. We think that extending ConGrs to responses from multiple models is a natural direction for future work. In fact, recent work such as [2] demonstrates that responses between different model classes also display homogeneity which could be exploited. One possible approach is replacing the *global* alignment that we use with *local* alignment techniques. While global alignment seeks to find optimal alignments that span entire responses, local alignment seeks to find independent matching segments across responses. We added a note about this in the conclusion of our paper, thank you for this suggestion.
>
> [1]: Predicting vs. acting: A trade-off between world modeling & agent modeling. Li et al., 2024.
>
> [2] Artificial Hivemind: The Open-Ended Homogeneity of Language Models (and Beyond), Jiang et al., NeurIPS 2025.
>
>
> ***Question 2**: How are hyperparameters selected? Do they require task-specific or model-specific tuning?*
>
> As we note in Section 4.1, the hyperparameters for both of our decoding methods reflect user preferences rather than task-specific tuning. For instance, the consensus decoding threshold parameter $\tau$ controls the tradeoff between factual precision and recall. If the user has a low tolerance for factual errors, they can select a higher value of $\tau$ and vice versa. Similarly, the hyperparameter $\kappa$ for guided self-verification controls how often a certain region of high-divergence is scrutinized. Figure 5 shows that our method achieves the best trade-off between factuality and informativeness at nearly all parameter settings for long-form factuality.

---

> > ### Comment · Area_Chair_qmu1 · 2025-11-22
> >
> > Hi Reviewer,
> >
> > The authors have submitted their responses to your reviews. Please take a look and let the authors know if you have any further questions or concerns. Thank you again for your contributions to ICLR!
> >
> > Best regards,
> > AC

---

### Official Review · Reviewer_rjPs · 2025-11-01

**Soundness:** 2
**Presentation:** 3
**Contribution:** 2
**Rating:** 4
**Confidence:** 3

**Summary:**

This paper proposes Consensus Graphs (CONGRS), a graph-based framework to synthesize multiple language models (LM) responses sampled for the same prompt. The authors adapt a multiple sequence alignment (MSA) algorithm (Needleman–Wunsch) from bioinformatics to align lexical sequences across responses, producing a directed acyclic graph (DAG) that encodes consensus and disagreement spans. Two decoding schemes are introduced:

Consensus Decoding, which aggregates high-agreement nodes to enhance factuality.

Guided Self-Verification, which targets high-disagreement regions to refine reasoning accuracy.

Experiments on factual generation (biography, PopQA), refusal tasks (HALoGEN), and reasoning tasks (MATH, AIME) show improved factual precision, abstention rate, and reasoning performance compared to several baselines.

**Strengths:**

The paper explores how to exploit response variation across LM samples as a reliability signal. This direction aligns with the interest in epistemic calibration, self-verification, and uncertainty-aware decoding for LMs.

The proposed pipeline (Sample to Align to Synthesize) is conceptually clear, and the DAG representation provides an interpretable way to identify consensus and conflict regions across responses.

Compared to prior LM-judge-heavy aggregation methods (e.g., UAD, ASC), the approach achieves similar or better performance with fewer tokens, which makes it practically appealing.

**Weaknesses:**

The core mechanism of using MSA (Needleman–Wunsch) to align token sequences and representing them as a DAG is a direct adaptation of standard sequence alignment and partial-order graph techniques.
The "bioinformatics inspiration" reads more like narrative packaging than like algorithmic innovation. The key insight (leveraging multi-sample alignment) could likely be realized without this heavy graph formalism.

The paper's framing ("node frequency as reliability") is conceptually appealing but not formally supported. There is no theoretical link between graph topology and uncertainty.

Constructing a lexical DAG across m responses of length L likely has high cost. The paper does not analyze runtime, memory, or trade-offs for larger-scale reasoning settings.

**Questions:**

How does CONGRS compare to simpler token-level overlap or attention-based voting schemes in both cost and quality?

How does performance scale with m (number of samples) and L (response length)?

---

> ### Author Response · Authors · 2025-11-22
> **Author Response**
>
> Thanks for your review!
>
> ***Weakness 1**: The method is a direct adaptation of standard multiple-sequence alignment techniques. The key insight (leveraging multi-sample alignment) could likely be realized without this heavy graph formalism.*
>
> Other methods do leverage agreement between multiple samples without the graph formalism. Most closely related, atomic self-consistency [1] aggregates information across responses at the sentence level by using sentence embeddings. The naive LM consensus in our paper and Fine-Grained Self-Consistency [2] use an external LM-judge to directly combine common spans across responses. Our method outperforms all of these, showing that the graph structure is beneficial for downstream tasks.
>
> [1]: Atomic Self-Consistency for Better Long Form Generations, Thirukovalluru et al., EMNLP 2024.
>
> [2]: Integrate the Essence and Eliminate the Dross: Fine-Grained Self-Consistency for Free-Form Language Generation.
>
> ***Weakness 2**: The paper's framing ("node frequency as reliability") is conceptually appealing but not formally supported. There is no theoretical link between graph topology and uncertainty.*
>
> We agree that there is no theoretical link between node frequency and reliability, and we do not claim one in the paper. However, the relation between response frequency and answer reliability forms the basis of the many well-established empirically successful techniques mentioned in our response above, such as self-consistency [3] and its many long-form variants [1, 2, 4, among others] and LM-assisted claim-based response decomposition [5]. We compare to these methods and find that using our approach results in higher performance.
>
> [3]: Self-Consistency Improves Chain of Thought Reasoning in Language Models, Wang et al., ICLR 2023.
>
> [4]: Universal Self-Consistency for Large Language Model Generation, Chen et al., 2023.
>
> [5]: Graph-based Uncertainty Metrics for Long-form Language Model Outputs, Jiang et al., NeurIPS 2024.
>
> ***Weakness 3 / Question 2**: Constructing a lexical DAG across m responses of length L likely has high cost. The paper does not analyze runtime, memory, or trade-offs for larger-scale reasoning settings. How does performance scale with m (number of samples) and L (response length)?*
>
> The MSA algorithm to construct the lexical DAG is an iterative  polynomial time dynamic programming algorithm which has asymptotic time complexity of $O(mL^2)$ where $m$ is the number of responses and L is the maximum length of a response. In all of our settings, the time needed to construct the initial lexical DAG is under 1 second. We have added this information to Section 2 and Appendix B in the paper, where we provide details about the alignment step, and we have already reported end-to-end runtime for our method across all experimental settings in Table 6 of Appendix G.
>
> ***Question 1**: How does CONGRS compare to simpler token-level overlap or attention-based voting schemes in both cost and quality?*
>
> In Section 4 of the paper, we compare our method to Minimum Bayes Risk (MBR) decoding, which uses lexical comparisons to select the response that is most similar to the other responses in the set (with similarity measured by BERTScore). We also have run a new comparison using Unigram Consistency Scores (UCS) [6], which performs a similar selection method to MBR (with unigram overlap as the similarity measure). We report the results below for Qwen 2.5 72B on biography generation below. Our method out-performs both methods in terms of factual precision, number of supported claims, and number of unsupported claims. In addition, while these other methods don’t require the costs of an external LM, they are limited to response selection rather than synthesis. We will incorporate these results into the paper as well.
>
> | Method | FActScore (&uarr;) | # supported claims (&uarr;) | # unsupported claims (&darr;) | response ratio |
> |-----|---:|--:|--:|--:|
> | MBR | 0.69 | 19.58 | 8.41 | 0.97 |
> | UCS | 0.68 | 18.46 | 8.40 | 0.98 |
> | ConGrs (tau = 0.3) | 0.70 | 20.81 | 6.97 | 0.93 |
> | ConGrs (tau = 0.5) | 0.77 | 17.48 | 3.48 | 0.85 |
>
> [6] Lightweight reranking for language model generations, Jain et al., ACL 2024.

---

> > ### Comment · Area_Chair_qmu1 · 2025-11-22
> >
> > Hi Reviewer,
> >
> > The authors have submitted their responses to your reviews. Please take a look and let the authors know if you have any further questions or concerns. Thank you again for your contributions to ICLR!
> >
> > Best regards,
> > AC

---

### Author Response · Authors · 2025-12-03
**Author Final Remarks**

Our paper introduces a flexible inference-time method to synthesize information from across multiple LLM responses to the same prompt. We construct a graph structure to capture response similarities and differences, which we then use to generate improved LLM responses in multiple domains, like hallucination reduction, abstaining from unanswerable questions, and reasoning. We thank the reviewers for their valuable comments. Reviewers find that our method achieves high performance across a wide variety of experimental settings (VmdJ, EiKb, 4yjc). They also find our approach novel (EiKb) and conceptually clear (rjPs, 4yjc).

Although reviewers did not respond during the shortened discussion period, we believe we have addressed their main concerns, as summarized below.

**Comparison to additional methods (Reviewers rjPs, VmdJ, 4yjc):**

Several reviewers asked us to compare our approach to alternative methods, including token-level overlap methods and self-consistency variants. First, our method can combine variable-length spans from multiple responses more flexibly than existing self-consistency variants (which either select whole responses or combine whole sentences across responses). Second, we ran additional experiments to compare to Universal Self-Consistency, Fine-Grained Self-Consistency, and Unigram Consistency Score. Our method outperforms all these methods, still achieving the Pareto-optimal tradeoff between factuality and informativeness. The paper shows that our method also outperforms Minimum Bayes Risk decoding and Atomic Self-Consistency as well.

**Impact of the LLM judge on performance (Reviewer EiKb):**

Reviewer EiKb raised concerns about our approach’s dependence on gpt-4.1-mini as an LLM judge for two steps: consensus graph construction and fixing grammatical errors in draft responses. We ran an experiment with Qwen-3-8b as a small, open LLM judge, which resulted in extremely similar performance. This shows that our method is robust to the choice of which particular LLM judge is used. We also expanded by 4x our manual validation of the final error fixing step, and we find that the external LLM judge rarely substantively changes the information in the draft response (< 0.5% of examined claims are changed or dropped by the judge).

**Cost-effectiveness of our method (Reviewers rjPs, VmdJ, 4yjc):**

Several reviewers raised concerns about the efficiency of our method. Although our method is inherently less efficient than methods that do not require a LLM judge, our method demonstrates consistently superior performance across three diverse settings (hallucination detection, abstention, and reasoning). Moreover, Table 2 of the paper shows that our method uses 82% fewer tokens than an alternative method that also uses LLM judges. The trade-off is clear: methods that do not use LLM judges are more efficient, but their performance is limited. To achieve better performance, LLM judges can be used to synthesize across multiple responses, where our method is the most efficient.

**Overall, we have addressed the primary weaknesses raised by the reviewers by running additional experiments. Our approach is on the Pareto frontier of efficiency and performance, and it also outperforms all additional baselines.**

---

### Meta-Review · Area_Chair_WtmQ · 2026-01-11

**Summary:**

The paper introduces a graph-based method to synthesize multiple LLM responses to given prompt. Three of the reviewers are mildly negative about the paper and one of them is mildly positive. The reviewers highlight a number of concerns which, even after taking into account the rebuttal, do not allow me to recommend acceptance.

**Reviewer Concerns:**

The concerns regarding the significance of the algorithmic contribution, lack of theoretical grounding, and generalizability of the framework are outstanding.

**Reviewer Scores:**

I do not think the reviewers would have significantly changed their scores.

---

### Decision · Program_Chairs · 2026-01-26

Reject